# Estimation of Secondary PM[2.5] in China and the United States using a Multi-Tracer Approach

**Haoran Zhang [1], Nan Li [1, *], Keqin Tang [1], Hong Liao [1, *], Chong Shi [2, 3], Cheng Huang [4], Hongli Wang [4], Song Guo [5], Min Hu [5], Xinlei Ge [1], Mindong Chen [1], Zhenxin Liu [1], Huan Yu [6], Jianlin Hu [1]**

[1] Jiangsu Key Laboratory of Atmospheric Environment Monitoring and Pollution Control, Jiangsu Collaborative Innovation Center of Atmospheric Environment and Equipment Technology, School of Environmental Science and Engineering, Nanjing University of Information Science & Technology, Nanjing, 210044, China

[2] National Institute for Environmental Studies,Center for Global Environmental Research,Tsukuba,Ibaraki,Japan

[3] Institute of Remote Sensing and Digital Earth, Chinese Academy of Sciences, Beijing, 100094, China

[4] State Environmental Protection Key Laboratory of Formation and Prevention of the Urban Air Pollution Complex, Shanghai Academy of Environmental Sciences, Shanghai, 200233, China

[5] College of Environmental Sciences and Engineering, Peking University, Beijing, 100871, China

[6] Department of Atmospheric Science, School of Environmental Studies, China University of Geosciences, Wuhan, 430074, China

[*] *Correspondence to:*

*Nan Li, linan@nuist.edu.cn and Hong Liao, hongliao@nuist.edu.cn*

**Abstract:**

PM$_{2.5}$, generated via both direct emissions and secondary formations, can have varying environmental impacts due to different physical and chemical properties of its components. However, traditional methods to quantify different PM$_{2.5}$ components are often based on online observations or lab analyses, which are generally high economic cost and labor-intensive. Chemical transport model (CTM) is another useful tool to reveal the composition characteristics of PM$_{2.5}$ but with high requirement of computation cost. In this study, we develop a new method, named multi-tracer estimation algorithm (MTEA), to identify the primary and secondary components from routine observation of PM$_{2.5}$. By comparing with the long-term and short-term measurements of aerosol chemical components in China and the United States, MTEA is proved to be able to successfully capture the magnitude and variation of the primary PM$_{2.5}$ (PPM) and secondary PM$_{2.5}$ (SPM). Meanwhile, our model poses a good agreement with the reanalysis dataset from one of the most advanced CTMs in China as well. Applying MTEA to China national air quality network, we find that 1) SPM accounts for 63.5% of PM$_{2.5}$ in southern cities of China averaged for 2014-2018, while in the North the proportion drops to 57.1%, and at the same time the secondary proportion in regional background regions is ~19% higher than that in populous regions; 2) the summertime secondary PM$_{2.5}$ proportion presents a slight but consistent increasing trend (from 58.5% to 59.2%) in most populous cities, mainly because of the recent increase in O$_3$ pollution in China; 3) the secondary PM$_{2.5}$ proportion in Beijing significantly increases by 34% during the COVID-19 lockdown, which might be the main reason of the observed unexpected PM pollution in this special period; and at least, 4) SPM and O$_3$ show similar positive correlations in the BTH and YRD regions, but the correlations between total PM$_{2.5}$ and O$_3$ in these two regions are quite different as PPM levels determine. In general, MTEA is a promising tool for efficiently estimating PPM and SPM, and has huge potential for future PM mitigation.

**Keywords:** PM$_{2.5}$; Primary and secondary sources; Air quality; O$_3$; COVID-19

# 1 Introduction

Fine particulate matter (PM$_{2.5}$, aerodynamic diameter less than 2.5 μm) can be categorized into primary and secondary PM$_{2.5}$ according to its formation processes. Primary PM$_{2.5}$ (PPM), including primary organic aerosol (POA), elemental carbon (EC), sea salt and mineral dust, is the product of direct emission from combustion of fossil/biomass fuel, dust blowing and sea spray. Secondary PM$_{2.5}$ (SPM) mainly generates from the further oxidation of gaseous precursors emitted by anthropogenic and biogenic activities (Zhu et al., 2018; Wang et al., 2019). SPM consists of secondary organic aerosol (SOA) and secondary inorganic aerosol (SIA, including sulfate, nitrate and ammonium). The primary and secondary components of PM$_{2.5}$ have different environmental impacts on air quality, human health and climate change. For example, as a typical PPM, EC can severely reduce atmospheric visibility and greatly influence weather and climate due to its strong absorption of solar radiation (Bond et al., 2013; IPCC, 2013; Mao et al., 2017). Sulfate, a critical hygroscopic component of secondary PM$_{2.5}$ (SPM), can be fast formed under high relative humidity conditions and further leads to grievous air pollution (Cheng et al., 2016; Guo et al., 2014; Quan et al., 2015). Furthermore, the sulfate and other hygroscopic PM$_{2.5}$ have considerable influences on climate change mostly by changing cloud properties (Leng et al., 2013; von Schneidemesser et al., 2015). In addition, different PM$_{2.5}$ components also have various deleterious impacts on human health for their toxicities (Hu et al., 2017; Khan et al., 2016; Maji et al., 2018).

To understand the severe PM$_{2.5}$ pollution characteristics in China over the past several years (An et al., 2019; Song et al., 2017; Yang et al., 2016), many observational studies have been conducted on PM$_{2.5}$ components. The basic methods of these studies are offline laboratory analysis and online instrument measurement such as aerosol mass spectrometer (AMS). The observational studies are crucial to exactly identify the aerosol chemical compositions. For offline approach, it is the most widely used method (Ming et al., 2017; Tang et al., 2017; Tao et al., 2017; Dai et al., 2018; Gao et al., 2018; Liu et al., 2018a; Wang et al., 2018; Zhang et al., 2018; Xu et al., 2019; Yu et al., 2019) and is successfully applied to investigate the inter-annual variations of different aerosol chemical species (Ding et al., 2019; Liu et al., 2018b). In terms of online approach, AMS is the state-of-the-art method for analyzing different chemical species with high time resolution, which has great application value in diagnosing the causes of haze events in China over the past decade (Huang et al., 2014b; Quan et al., 2015; Guo et al., 2014; Yang et al., 2021; Gao et al., 2021; Hu et al., 2021;

Zhang et al., 2022).

Nevertheless, both the online and offline measurements require a high level of manpower and economic cost, and for this reason, these methods are expensive and rarely applied in large-scale regions or long-term periods.

Chemical transport model (CTM) is another useful tool to identify the composition characteristics of $PM_{2.5}$. The simulation predicted by CTM is featured as high spatio-temporal resolution (Geng et al., 2021). Meanwhile, it also provides vertical profiles of diverse chemical species (Ding et al., 2016). However, the CTM results are largely dependent on external inputs such as emission inventories, boundary conditions, initial conditions, etc. The internal parameterizations of itself significantly influence the final model results as well (Huang et al., 2021), which leads to uncertainty in the simulated $PM_{2.5}$ and its composition. In addition, the burden of high requirement in computational cost and storage also makes CTM hard to universally use.

In this study, we develop a novel method, Multi-Tracer Estimation Algorithm (MTEA), with the aim of distinguishing the primary and secondary compositions of $PM_{2.5}$ from routine observation of $PM_{2.5}$ concentration. Different from traditional CTMs, MTEA proposed by this study is based on statistical assumption and works in a more convenient way. This algorithm and its application are tested in China and the United States. In Section 2, we introduce the structure and principle of MTEA. In Section 3, we evaluate the MTEA results comparing with three $PM_{2.5}$ composition data sets, (1) short-term measurements in 16 cities in China from 2012 to 2016 reported by previous studies, (2) continuous long-term measurements in Beijing and Shanghai from 2014 to 2018, and (3) IMPROVE network in the United States during 2014 and 2018. Additionally, we also compare MTEA model with one of the most advanced datasets from CTM in China. Subsequently, in Section 4 we investigate the spatio-temporal characteristics of PPM and SPM concentrations in China, explain the unexpected haze event in several cities of China during the COVID-19 lockdown and discuss the complicated correlation between PM and $O_3$. This study is different from previous works as follows: (1) we develop an efficient approach to explore PPM and SPM with low economy-/technique-cost and computation burden, (2) we apply this approach to observation data from the MEE network, offering an unprecedented opportunity to quantify the $PM_{2.5}$ components on a large space and time scale.

## 2 Methodology

### 2.1 The Multi-Tracer Estimation Algorithm (MTEA)

In order to distinguish PPM and SPM efficiently from the observed $PM_{2.5}$, we develop a new approach, named Multi-Tracer Estimation Algorithm (MTEA). The multi-tracer (marked as X) is defined to represent multiple primary contributions to $PM_{2.5}$, mainly resulting from incomplete combustion of carbonaceous material and flying dust. We select the typical combustion product CO as one tracer to represent the combustion process, and the particles in coarse mode ($PM_{coarse}$, marked as PMC, $PMC = PM_{10} - PM_{2.5}$) as the other tracer to track flying dust. Then, we combine the CO and PMC to generate the multi-tracer X (Eq. 1), which can represent hybrid primary contributions to $PM_{2.5}$.

$$X = a * CO + b * PMC \quad (a + b = 100\%) \quad (1)$$

As shown in Eq. 1, we use $a$ and $b$ to quantify the relative contributions of combustion and dust process to PPM. Given that the complicated process such as the combustion from multiple sources is hard to represent via current routine CO observations, we avoid considering the correlation among these sources but focus on the relative weights of combustion process and flying dust. Meanwhile, the uncertainty resulting from the apportioning coefficient $a$ and $b$ will be further discussed in Section 4.5. The values of the coefficients depend on the ratio of emission intensities of POA+EC (combustion products) and fine mode dust, as shown in Eq. 2.

$$\frac{a}{b} = \frac{E_{OA} + E_{EC}}{E_{finedust}} = \frac{1.2E_{OC} + E_{EC}}{E_{PM2.5} - (1.2E_{OC} + E_{EC} + E_{SO4} + E_{NO3})} \quad (2)$$

where, $E_{OA}$, $E_{EC}$, $E_{finedust}$, $E_{OC}$, $E_{PM2.5}$, $E_{SO4}$ and $E_{NO3}$ represent the emissions of OA, EC, fine mode dust, OC, $PM_{2.5}$, sulfate and nitrate, respectively. We obtain anthropogenic $PM_{2.5}$, EC and OC emissions in China from Multi-resolution Emission Inventory for China (MEIC, *http://meicmodel.org/*, last access: 1 August 2021) developed by Tsinghua University (Li et al., 2017c). For the United States, we retrieve the emission data from the global inventory HTAP (*https://edgar.jrc.ec.europa.eu/htap_v2/index.php?SECURE=123*, last access: 1 August 2021). We further estimate POA emission using POC emission multiply by an empirical factor of 1.2 recommended in literature (Seinfeld and Pandis, 2006), and quantify sulfate and nitrate emissions using $PM_{2.5}$ emission multiply by an investigative coefficient of 0.1 (Zhang 2019). However, this investigative coefficient for quantifying primary sulfate and nitrate emissions might be relatively higher compared to empirical coefficients (0.01-0.05) used in previous simulation studies. We evaluated the potential effect of the coefficient, by

conducting a set of comparative simulation with the coefficient of 0.03, and found that the final estimated SPM was not sensitive to this coefficient (Table S1). Thus we concluded that the uncertainty of primary sulfate and nitrate emissions did not significantly influence the final estimation of MTEA model. For other uncertainties of X which are dependent on emission intensities or tracer concentrations, we would conduct discussions in the later Section 4.5. Coefficient $b$ is aimed at reflecting the activity intensity of fine mode dust by counting its emissions. However, MEIC does not directly provide fine mode dust emissions. It is included in the emissions of total $PM_{2.5}$ (Li et al., 2017b). Thus we inferred the fine mode dust emission by deducting the emissions of EC, POA, sulfate and nitrate from the $PM_{2.5}$ emissions. Based on Eq. 2, we establish a dynamic "$a$-$b$ value" database, which can reflect the specific changes of $PM_{2.5}$ sources in terms of different years, seasons, hours, and different regions.

With the help of the multi-tracer X, we can describe secondary $PM_{2.5}$ as follows:

$$SPM = PM_{2.5} - PPM \qquad (3)$$

$$= PM_{2.5} - \frac{PPM}{X} \times X \qquad (4)$$

Here, $PM_{2.5}$ is the observed $PM_{2.5}$ concentration, and the multi-tracer X can be calculated from the observed CO, $PM_{2.5}$ and $PM_{10}$ concentrations. The original concentrations of CO, $PM_{2.5}$ and $PM_{10}$ are normalized to avoid the influences of their initial levels. To calculate SPM, the key step is to find out the target ratio of PPM/X. In the MTEA method, we give the PPM/X ratio a reasonable range (a range from 0 to 400 is used in this work) and then scan the ratio with an interval of 1. For more precise results, a smaller scanning step can be applied while it may take larger calculation cost. As a result, each varying ratio may obtain a series of SPM, along with a coefficient of determination ($R^2$) between SPM and X (Fig. S1). If we assume that PPM and SPM came from different sources or processes, then the appropriate PPM/X ratio should be the one that corresponds to weak correlation between SPM and X-tracer. To better understand the principle of the MTEA approach, we show the flow chart in Fig. 1. We also provide the MTEA software package and input data sets at *http://nuistairquality.com/m_tea* (last access: 1 August 2021).

The MTEA approach makes some improvement based on the similar principle and assumptions with the modified EC-tracer method developed by Hu et al. (2012). They estimated primary and secondary organic carbon (marked as POC and SOC) concentrations

by adopting a proper POC/EC ratio when SOC correlated with EC worst. However, this assumption may be too hard to exist in the real atmosphere. Therefore in the MTEA approach, we take a range of proper ratios of PPM/X when SPM correlates with X-tracer non-significantly (with $p$-value greater than 0.05). As a result, the calculated SPM concentration for each case is a range (Table S2). We employed the concentration ranges to represent the severity of secondary pollution and discussed its uncertainties in the following discussions. While for quantitative calculation, the mean values of the concentration ranges stand for the final estimation.

## 2.2 PM$_{2.5}$ measurements

### 2.2.1 PM$_{2.5}$ concentration measurements from the MEE network in China

Focus on the PM$_{2.5}$ pollution in China, MEE set up a comprehensive air quality monitoring network for consistently accessing hourly concentrations of PM$_{2.5}$ as well as SO$_2$, NO$_2$, CO, O$_3$ and PM$_{10}$ since 2013. This network is the most advanced monitoring network currently in China. In this study, we obtained surface observations of hourly PM$_{2.5}$, PM$_{10}$, CO and O$_3$ at 334 national monitoring sites in 50 cities from 2014 to 2018 from the MEE public website (*http://106.37.208.233:20035/*, last access: 1 August 2021). 31 among the 50 cities are provincial capital cities, employed to represent populous cities, while the rest 19 relatively small cities are categorized as regional background cities (Table S3). The mean PM$_{2.5}$ concentration of each regional background city is less than 35.0 μg·m$^{-3}$ (National Ambient Air Quality Standard level II of China, NAAQS) except for Guyuan, indicating that they are slightly impacted by anthropogenic activities. By comparing populous cities with regional background cities, we could reveal the discrepancy in PPM and SPM among those regions which suffer from different levels of PM$_{2.5}$ pollution. Geographical distribution of those populous and regional background cities is shown in Fig. 2a.

Recently, the Chinese government carried out a series of control policies, such as elimination of backward industry, desulfurization and denitration of flue gas, as well as restriction on motor vehicles (Tang et al., 2019; Wu et al., 2017). Consequently, the concentrations of the major gaseous and particle pollutants have been decreased year by year (Zhai et al., 2019; Shen et al., 2020) . Take PM$_{2.5}$ as an example, previous studies revealed that annual mean PM$_{2.5}$ decreased by 30-50% across China during the period of 2013-2018.

### 2.2.2 PM2.5 composition measurements in China

Numerous studies focused on the aerosol chemical composition in China employed offline filter-based observations coupled with laboratory analysis to obtain detailed information of PM2.5 compositions. For directly comparing the estimated PPM/SPM with the measured ones in China, we made an evaluation via two long-term time series in-situ measurements in Beijing (Peking University, PKU) and Shanghai (Shanghai Academy of Environment Sciences, SAES) during 2014-2018 (Huang et al., 2019; Tan et al., 2018). The chemical compositions of measurements include ions ($NH_4^+$, $Na^+$, $K^+$ $Mg^{2+}$, $Ca^{2+}$, $SO_4^{2-}$, $NO_3^-$, $Cl^-$, by ion chromatography), elements (Al, Si, Ti, Ca, Ti, Mn, etc., through X-ray fluorescence spectrometry), and carbonaceous components (EC and organic carbon, using the thermal-optical transmittance carbon analyzer). After accessing the chemical compositions, we categorized them into PPM and SPM for further evaluation. Specifically, SOA was roughly identified from OM by EC-tracer model (Ge et al., 2017). SPM concentrations were calculated via summing $SO_4^{2-}$, $NO_3^-$, $NH_4^+$ and SOA concentrations. Then PPM could be calculated though deducting SPM from PM2.5.

In addition, we investigated observation-based PM2.5 component analyses in 16 cities of China during 2012-2016 from 32 published studies. This survey offered an opportunity to compare the estimation by MTEA with the past measurements in the terms of the secondary fraction of PM2.5. SPM concentrations in literature were roughly estimated by multiplying OM from 0.5 because of the limit of data source. Meanwhile, it is noted that the factor which converts OC to OM is dependent on the definition of each observation study itself.

### 2.2.3 PM2.5 composition measurements from IMPROVE network in the United States

The Interagency Monitoring of Protected Visual Environments (IMPROVE) aerosol network has continuous records of $PM_{10}$, PM2.5 and its chemical speciation in the United States since 1987. The specific aerosol chemical compositions include ammonium sulfate, ammonium nitrate, organic/elemental carbon and soil/mineral dust. The categorization for PPM and SPM in IMPROVE dataset is similar to the process in Section 2.2.2. The only difference is that SPM concentration is the sum of ammonium sulfate, ammonium nitrate and SOA. More detailed descriptions about IMPORVE are available at *http://vista.cira.colostate.edu/Improve/* (last access: 1 August 2021). Here we extracted the measurements at 104 valid sites in the United States from 2014 to 2018 for the evaluation of

MTEA. The spatial distribution of IMPROVE sites used in this work is shown in Fig. 2b. It is noted that IMPROVE program only provides a single aerosol component profile every three days. We lowered the time resolution into the monthly average for further evaluation. However, CO is excluded in IMPROVE program. We therefore adopted the Kriging interpolation of CO data based on the hourly archives from the United States EPA (*https://www.epa.gov/outdoor-air-quality-data*, last access: 1 August 2021) as an alternative for model input when running the MTEA.

### 2.3 PPM and SPM estimated by CTM

Apart from evaluating PPM and SPM with various composition measurements, we also compared MTEA estimation with CTM results. Here we utilized the $PM_{2.5}$ composition gridded dataset with a spatial resolution of 10 km×10 km developed by Tsinghua University for further comparisons. This dataset is named Tracking Air Pollution in China (TAP, available at http://tapdata.org.cn/, last access 15 Mar 2022) (Geng et al., 2021; Geng et al., 2017). TAP is directly calculated by Community Multiscale Air Quality (CMAQ) model. In terms of methodology, based on machine learning algorithms, TAP integrates surface measurements, satellite remote sensing retrievals, emission inventories (MEIC) with CMAQ simulations. Moreover, it is also constrained by ground aerosol composition measurements. We collected the monthly mean concentrations of aerosol species during 2014-2018 from TAP, including $SO_4^{2-}$, $NO_3^-$, $NH_4^+$, OM, BC and total $PM_{2.5}$. SOA was further calculated from OM by EC-tracer model (Ge et al., 2017). SPM concentrations were inferred by summing $SO_4^{2-}$, $NO_3^-$, $NH_4^+$ and SOA. PPM concentrations were then obtained via deducting SPM from $PM_{2.5}$.

### 3 Model evaluation

### 3.1 Evaluation in China

### 3.1.1 Comparison with continuous long-term measurements in Beijing and Shanghai

We compared the MTEA results with the two sets of long-term in-situ measurements in Beijing and Shanghai, China, and show the evaluations in Fig. 3. Reduced major axis (RMA) regression was applied for fitting the data. Given the discrepancy in $PM_{2.5}$ concentrations between in-situ measurements of a single site and multiple MEE national sites, we firstly

preprocessed the data for further evaluation. In data preprocessing, we removed the in-situ daily measurements whose value was over 30 $\mu g \cdot m^{-3}$ higher than the city average (from MEE).

The comparisons between the estimated and observed PPM in the two cities are given in Fig. 3a and 3c. The correlation coefficient r for predicted PPM versus observed PPM is 0.85 in Beijing and 0.87 in Shanghai. The slope of regression is 1.29 in Beijing and 0.73 in Shanghai, which indicates an overestimation (NMB=32%) or underestimation (NMB= – 9%) in these two cities. In terms of SPM, the regression line in Shanghai is quite close to the 1:1 ratio line (s=1.13, d= – 2.3), and its statistical correlation is up to 0.89. The estimated SPM in Beijing also shows a high correlation with the observed ones, with its r value exceeding 0.80, though the fitting formula indicates an underestimation of 27%. The discrepancies can be explained by the fact that the observations of primary emission tracers and $PM_{2.5}$ are obtained from different sites. Specifically, the CO and PMC observations are obtained from 12 monitoring MEE sites in Beijing, while the $PM_{2.5}$ component measurements are from single spot at PKU which is away from crowded streets (Tan et al., 2018). The MTEA predictions based on the data from MEE sites located at high-emitting densities district may propose a quite overestimation on PPM concentrations.

Overall, MTEA model performed satisfactorily in case of the comparison with the long-term in-situ measurements in Beijing and Shanghai. Nearly all the dots are located at the range between 2:1 ratio and 1:2 ratio. It is believed that our model is able to capture the magnitudes and variations of the PPM and SPM. The comparison about the estimated and the observed inter-annual variations in PPM and SPM would be further discussed in the following texts (Sect. 4.2.2).

### 3.1.2 Comparison with various short-term measurements

To evaluate the reliability of the MTEA approach, we also conducted a literature review for collecting a variety of observation-based $PM_{2.5}$ component analyses in 16 cities of China during 2012-2016 (Chen et al., 2016; Du et al., 2017; Cui et al., 2015; Dai et al., 2018; Gao et al., 2018; Huang et al., 2014a; Huang et al., 2014b; Huang et al., 2017; Jiang et al., 2017; Li et al., 2016; Li et al., 2017a; Lin et al., 2016; Liu et al., 2017; Liu et al., 2014; Liu et al., 2018a; Liu et al., 2018b; Ming et al., 2017; Niu et al., 2016; Tan et al., 2016; Tang et al., 2017; Tao et al., 2017; Tao et al., 2015; Tian et al., 2015; Wang et al., 2018; Wang et al., 2016a;

Wang et al., 2016b; Wu et al., 2016; Xu et al., 2019; Yu et al., 2019; Zhang et al., 2015; Zhang et al., 2018; Zhao et al., 2015). Most field measurements focused on regions in eastern China and on episodes during the winter. We listed the concentrations of observed $PM_{2.5}$, $SO_4^{2-}$, $NO_3^-$, $NH_4^+$, and SOA from these studies in Table S4. It should be noted that there may be inconsistencies in the observation due to different sampling locations, observational time and analytical instruments in each study.

The estimated PPM and SPM from MTEA show a reasonable agreement with the observation-based $PM_{2.5}$ component analyses in China. The MTEA estimated secondary proportions of $PM_{2.5}$ (i.e. secondary $PM_{2.5}$ / total $PM_{2.5}$) vary in a range of 41% to 67%, and are higher in eastern cities of China, consistent with the observational results. However, we find that there are still a few discrepancies between the estimated and observation-based results. For example, we overestimated the secondary proportions of $PM_{2.5}$ in cities such as Haikou, Lanzhou and Lhasa. Though all of them show a considerable overestimation of over 20%, the causes lead to this kind of bias may be quite different. In coastal city Haikou, we may attribute this discrepancy between MTEA and observation to the neglect of the contribution of sea salt aerosols. The $PM_{2.5}$ offline measurements in 2015 exhibited that the contribution of sea salt aerosols to ambient $PM_{2.5}$ mass concentration in Haikou is 3.6-8.3% (Liu et al., 2017). Secondly, the overestimation phenomenon in Lanzhou, which is a typical inland city located in northwestern China, can be explained by overlooking the contribution of natural dust to $PM_{2.5}$ speciation. Generally, both sea salt and natural dust are categorized into non-anthropogenic processes, and are not accounted for by anthropogenic emission inventory, resulting in the underestimation of representing primary process intensity. Finally, for Lhasa, the observation-based results which are derived from too few samplers also pose controversial comparison against MTEA model.

### 3.1.3 Comparison with the CTM simulation

In addition to evaluating our model via PPM and SPM measurements in China, we also provided a comparison between MTEA estimation and CTM simulation in 31 populous cities based on the monthly mean PM concentrations. As shown in Fig. 4 a-b, the correlation coefficient r for TAP versus MTEA is 0.86 in terms of PPM concentration and 0.91 in terms of SPM concentration, showing a strongly positive correlation between the two models. At the same time, both slopes (1.26 and 0.89) and intercepts (–3.7 μg m$^{-3}$ and 1.9 μg m$^{-3}$) of the

regression about PPM and SPM illustrate that most of the scattering spots distribute around 1:1 ratio line.

Moreover, we further compared the long-term varying trends between MTEA versus TAP in averaged PPM and SPM concentration of 31 populous cities (Fig. 4 c-d). Both of them exhibit a descending interannual trend in PPM concentration, with a rate of $-2.0$ μg m$^{-3}$ yr$^{-1}$ for MTEA and $-1.9$ μg m$^{-3}$ yr$^{-1}$ for TAP. In terms of SPM concentration, the decline rates are $-2.9$ μg m$^{-3}$ yr$^{-1}$ and $-2.8$ μg m$^{-3}$ yr$^{-1}$, respectively. Meanwhile, the statistical correlations between two interannual variations are 0.98 (PPM) and 0.99 (SPM), which are quite close to 1, showing a good agreement.

Thus, the comparisons about PPM/SPM concentration magnitudes and interannual variations between two kinds of models suggest that statistical model can infer similar estimation with traditional CTM. Meanwhile, it is again highlighted that our model is capable of capturing reasonable PPM and SPM concentrations. Furthermore, it is also shown that MTEA can track primary and secondary component of PM$_{2.5}$ by using proxy at a much lower cost when compared to traditional air quality model simulations.

### 3.2 Evaluation in the United States

Based on the chemical component measurements of IMPROVE network, we evaluated the performance of the MTEA model in the United States. Figure 5 presents the scatter plots of the evaluation results, with x-axis indicates the observed concentrations and the y-axis indicates the estimated concentrations. The validation was done in the form of temporal, spatial, as well as spatio-temporal. Each dot represents a monthly mean of either observed or estimated PM concentration.

Almost all of the dots are located in the region between the 2:1 and 1:2 dotted line, indicating that our model is capable of predicting the magnitudes of PPM/SPM in the United States. Based on correlation analysis, we find that the correlation coefficient r for PPM ranges from 0.69 (spatio-temporal validation) to 0.75 (temporal validation), while for SPM, the r is even up to 0.98 (temporal validation). The results reveal that the MTEA approach successfully captured the spatial and temporal variations of PPM and SPM in the United States.

The majority of dots are distributed around the 1:1 dotted line. Based on the fitting results, the slopes for regression lines vary from 1.12 (spatial validation) to 1.15 (temporal

validation) for PPM and from 0.92 (temporal validation) to 0.93 (spatio-temporal validation) for SPM.  In general, PPM and SPM show a slight overestimation and underestimation, respectively. The discrepancies may result from the influences of emission inventory. It is reported that the emissions of PMC and CO in the United States continuously declined over the past decade (https://www.statista.com/statistics/501298/volume-of-particulate-matter-2-5-emissions-us/, last access: 2 October 2021). Thus the coefficients a and b derived from HTAP global emission inventory in 2010 overestimate the contribution of primary emissions during the studying period. However, the impacts of emission are inevitable, and we will discuss the uncertainty of emission inventory in Sect. 4.5. In addition, the intercepts of these regression lines for both PPM and SPM are less than $\pm 0.1$ $\mu g \cdot m^{-3}$. The verification results strongly show that our model can reasonably reproduce the monthly averaged concentration of PPM and SPM in the United States.

## 4 Results and discussion

We used the MTEA approach and the MEE observation data to estimate PPM and SPM concentrations in China for the period of 2014-2018. The observations during severe haze events (top 10% CO and PMC polluted days) were excluded to avoid the influence of unfavorable meteorological conditions and extreme high primary emission cases. Unfavorable meteorological conditions are major causes for haze events. PPM under these unfavored meteorological conditions may have considerable high co-linear relationship with total $PM_{2.5}$. The concentration of SPM from complicated formation pathways is then underestimated. Therefore, we excluded these polluted days to focus more attention on general characteristics of PPM and SPM concentration.

### 4.1 Spatial distribution

Figure 6 shows spatial patterns of the MTEA estimated PPM and SPM concentrations over China averaged for the period of 2014-2018. 16 populous cities and 9 regional background cities in the north, and 15 populous cities and 10 regional background cities in the south (North-South is separated by the Qinling-Huaihe line) are involved in the following discussions.

In populous cities, the concentrations of both PPM and SPM in the north (5-year averaged 21.5 $\mu g \cdot m^{-3}$ for PPM and 26.6 $\mu g \cdot m^{-3}$ for SPM) are 15-43% higher than those in the

south (15.0 μg·m$^{-3}$ for PPM and 23.2 μg·m$^{-3}$ for SPM). The North-South difference is mainly caused by the higher energy consumption and consequent stronger pollutant emissions occurring in northern populous regions. Nevertheless, in background regions, the difference is relatively smaller for SPM. The SPM in the South (12.5 μg·m$^{-3}$) is only 1% higher than that in the North (12.4 μg·m$^{-3}$).

In terms of the secondary proportion of PM$_{2.5}$, the MTEA approach speculates it to be higher in southern regions (63.5%) than that in northern regions (57.1%). The result confirms the fact that atmospheric condition in the South is more favorable for secondary pollutant formation than it is in the North. In addition, the MTEA approach reasonably captures the difference of the secondary proportion of PM$_{2.5}$ between populous and regional background cities. As shown in Fig. 6e and 6f, the secondary proportions of PM$_{2.5}$ in regional background cities are 19% higher than those in populous cities, consistent with recent observational studies (Liu et al., 2018b). Secondary aerosols can affect a larger area than primary aerosols, mostly due to the diffusion of its gaseous precursors. Thus, for regional background regions, the role of secondary PM$_{2.5}$ tends to be more important, mainly caused by the transmitted secondary pollutants from surrounding populous regions.

**4.2 Temporal variation**

**4.2.1 Seasonal variation**

We compare seasonal mean concentrations of the MTEA estimated PPM and SPM in 31 populous cities and 19 regional background cities in Table 1. Both the concentrations of PPM and SPM are the highest in winter, with the seasonal mean concentration of 16.6 μg·m$^{-3}$ for PPM and 24.9 μg·m$^{-3}$ for SPM across China. This phenomenon can be mainly explained by adverse diffusion conditions, such as low boundary layer height and strong temperature inversion (Zhao et al., 2013), as well as fossil-fuel and biofuel usage for winter home heating (Zhang et al., 2009; Zhang and Cao, 2015). Summer is the least polluted season in the year, with the seasonal mean PPM is 10.2 μg·m$^{-3}$ and SPM is 15.8 μg·m$^{-3}$ nationwide, largely benefiting from the higher boundary layer (Guo et al., 2019) and abundant precipitations.

In terms of the secondary proportion of PM$_{2.5}$, we also compared the secondary contributions in different seasons and in the 50 different Chinese cities (Table 1). The MTEA approach estimates that the secondary proportion tends to be the lowest in fall, with seasonal mean value to be 56.1% nationwide, while for the other three seasons, the seasonal

proportions stay around 61%. At the same time, the seasonality of the secondary proportion varies among different regions. In the north of China, the secondary proportions are higher in spring and summer, which is attributed to the stronger atmospheric oxidizing capacity (AOC) in the warmer seasons. But in the south of China, the highest secondary proportions occur in winter, which is mainly explained by the tremendous pollutants (secondary particles and its gaseous precursors) transported from northern China in the presence of the monsoon.

### 4.2.2 Inter-annual variation

Figure 7 illustrates the inter-annual variations of the estimated PPM and SPM based on MTEA in the 31 populous cities and 19 regional background cities of China. We analyzed the MEE observational data during 2014-2018, but excluded the data in 2014 in the regional background regions due to data deficiencies in several cities.

The observed $PM_{2.5}$ concentrations in populous cities are continuously and significantly reduced since 2014, largely benefiting from a series of emission control measures led by the governments, such as "Action Plan on Prevention and Control of Air Pollution" (Chinese State Council, 2013). Using the MTEA approach, we find that both PPM and SPM are decreased simultaneously, at an annual decreasing rate of 1.9 $\mu g \cdot m^{-3} \cdot yr^{-1}$ and 2.7 $\mu g \cdot m^{-3} \cdot yr^{-1}$, respectively. Consequently, the secondary proportion of $PM_{2.5}$ remains relatively constant (56.4-58.5%). But it presents a consistent increase trend (from 58.5% to 59.2%) in summer during the studying period, which can be attributed to the continuing worsening $O_3$ pollution (Tang et al., 2022). However, for regional background cities, the MTEA approach reports different features of the $PM_{2.5}$ mitigation. The estimated SPM is considerably reduced by 1.1 $\mu g \cdot m^{-3}$ $yr^{-1}$ in regional background cities, while the PPM keeps nearly unchanged (decreasing rate is 0.2 $\mu g \cdot m^{-3} \cdot yr^{-1}$). This is because SPM in regional background cities is largely contributed by pollutants transport from surrounding populous regions, where the air quality is getting better resulting from the aforementioned emission controls. However, the PPM, mostly deriving from local sources, is rarely affected by those emission controls which do mostly focus on densely-populated and industrialized cities but not on background regions.

We discussed the inter-annual variations of PPM and SPM concentration on the basis of long-term in-situ observations in Beijing and Shanghai as well. As Fig. 8 shows, long-term measurements demonstrate a decline of total $PM_{2.5}$ by 4.0 $\mu g \cdot m^{-3}$ $yr^{-1}$ in Beijing (1.6 $\mu g \cdot m^{-3}$ $yr^{-1}$ for PPM and 2.4 $\mu g \cdot m^{-3}$ $yr^{-1}$for SPM) and by 3.9 $\mu g \cdot m^{-3}$ $yr^{-1}$ in Shanghai (1.7 $\mu g \cdot m^{-3}$ $yr^{-1}$

for PM and 2.2 μg·m$^{-3}$ yr$^{-1}$for SPM). The observed secondary proportion of PM$_{2.5}$ shows a slight decrease of -0.4% yr$^{-1}$ in Beijing, but a small increase of 0.8% yr$^{-1}$ in Shanghai. Applying the MTEA model to this case, we are delighted to find that our model not only successfully reproduces the consistent decreasing trends of PPM and SPM in Beijing and Shanghai (correlation coefficient r of observation versus estimation ranges from 0.83 to 0.89), but also captures the different trends in secondary proportions of PM$_{2.5}$ in the two cities (-0.6% yr$^{-1}$ in Beijing and 0.3% yr$^{-1}$ in Shanghai).

### 4.3 Application during the COVID-19 lockdown

To curb the spread of the novel Coronavirus Disease 2019 (COVID-19) pandemic, China conducted the entire city's lockdown first in Wuhan, Hubei on January 23, 2020. Other provinces gradually implemented this restriction in the following three weeks (Le et al., 2020). The lockdown greatly limited the traffic and outdoor activities, which directly reduced the emissions of primary pollutants (Huang et al., 2020). Through analyzing the MEE monitoring data before (1~23 Jan 2020) and during (24-Jan ~ 17-Feb 2020) the nationwide lockdown (Fig. 9 and Fig. S2), we show that the national mean NO$_2$, PM$_{2.5}$ and CO concentrations were decreased by 56%, 30%, and 24%, respectively, while O$_3$ posed an increase (34%) in general which would promote the AOC efficiently. However, the surface monitoring network still observed an unexpected PM$_{2.5}$ pollution in cities over BTH region during the lockdown. Especially in Beijing, the mean PM$_{2.5}$ concentration was increased by ~100% compared to its averaged value (41 μg·m$^{-3}$) before the nationwide lockdown.

To explore this unexpected air pollution, we find that the enhanced secondary pollution could be the major factor, which even offset the reduction of primary emissions in the BTH region during the lockdown. With the help of MTEA, we tracked variations of the secondary proportions of PM$_{2.5}$ in East China before and during the COVID-19 lockdown (Fig. 9 d-f). The specific emission reductions owing to the national lockdown were derived from Huang et al. (2020). Based on the bottom-up dynamic estimation, provincial emissions of CO, NO$_x$, SO$_2$, VOC, PM$_{2.5}$, BC and OC decreased by 13-41%, 29-57%, 15-42%, 28-46%, 9-34%, 13-54%, and 3-42%, respectively during the lockdown period. The secondary proportions in the BTH region show an evident increase, at the level of 7%-34%, which highlights the importance of the secondary formation during the lockdown. Our result is consistent with recent observation and simulation studies (Chang et al., 2020; Huang et al., 2020; Le et al.,

2020), which suggested that the reduced $NO_2$ resulted in $O_3$ enhancement, further increasing the AOC and facilitating secondary aerosol formation. In addition, another cause of the air pollution is the unfavorable atmospheric diffusion conditions. CO, a nonreactive pollutant, was increased by 22% in Beijing during the lockdown even under considerable reduction on its emission.

For other regions of China, the MTEA approach suggests the secondary proportions of $PM_{2.5}$ to be increased by 20% over the YRD region, but to be decreased by 32% over Central China. Although $O_3$ and AOC had enhanced in all these regions, the unprecedented reductions on precursors ultimately resulted in a net drop in secondary pollution.

### 4.4 Correlation analysis with $O_3$

$PM_{2.5}$ and $O_3$ are closely correlated with each other. One reason is that $PM_{2.5}$ and $O_3$ have similar precursors, i.e. $NO_x$ and VOCs. Besides, $PM_{2.5}$ can impact $O_3$ formation through adjusting radiation balance (Li et al., 2018) and affecting radical level via aerosol chemistry (Li et al., 2019). There is therefore a complicated interaction between $PM_{2.5}$ and $O_3$. Our study utilized the MTEA approach for exploring the relationship between PM versus $O_3$ from the perspective of exploring the statistical correlation.

Figure S3 illustrates the hourly correlations between the estimated SPM versus the observed $O_3$ averaged for 31 populous cities in China (cities which failed to pass the significant test were excluded) in summer. In general, SPM and $O_3$ show a nationwide positive relationship, especially during the afternoon (14:00~18:00, r up to 0.56). This phenomenon might be explained that productions of both $O_3$ and SPM are simultaneously affected by AOC; thus the higher correlation tends to occur at time of stronger AOC. Moreover, the hourly correlations between SPM and $O_3$ are higher than that between PPM and $O_3$ throughout the day, suggesting that secondary oxidation processes may be well captured by the MTEA method.

A series of recent studies have focused on the correlation between $PM_{2.5}$ and $O_3$, and many of them agreed that the correlation varies greatly in different regions of China. Specifically, the statistical correlation is stronger positive in southern cities compared to that in northern cities (Chu et al., 2020). Because of this significant difference, a question raises: is the difference mostly caused by PPM, or SPM, or both of them? To address this question, we compare the correlations between daily PPM, SPM and total $PM_{2.5}$ versus $O_3$ in Beijing-

Tianjin-Hebei (BTH) and Yangtze River Delta (YRD) region during the studying period, with the help of META approach. The $O_3$ diurnal formation regime can be destroyed because of the suppressed radiative condition under precipitation. The local $O_3$ concentration level is mainly dominated by background fields. Here we would like to focus our attention on the secondary formation relationship between daily $PM_{2.5}$ and $O_3$. Therefore the cases when precipitation took place were removed to avoid the cleaning impacts of wet deposition on MDA8 (maximum daily 8-h average) $O_3$ concentrations. Precipitation data is based on the ERA5 reanalysis database from the European Centre for Medium-Range Weather Forecasts (ECMWF, *https://www.ecmwf.int/*, last access, 1 August 2021).

As shown in Fig. 10, the correlations between total $PM_{2.5}$ and $O_3$ are positive and are stronger in YRD (r=0.14) than in BTH (r=0.09). However, compared with total $PM_{2.5}$, the correlations between SPM and $O_3$ are much stronger (r=0.21-0.24) and show minor regional differences, but for PPM, its correlation with $O_3$ is not significant (p-value>0.05) in both regions. The higher correlation between SPM and $O_3$ is mostly because both of them are secondary oxidation products. Higher ambient $O_3$ concentration indicates stronger AOC, and further leads to more SPM generation. However, for PPM, its effect on $O_3$ is mainly to inhibit the production of $O_3$ via adjusting radiation balance and affecting radical level. Hence, we suggest that the regional differences in the correlation between total $PM_{2.5}$ and $O_3$ are mainly caused by the different PPM levels in BTH and YRD regions.

**4.5 Uncertainties**

Based on the previous evaluation and discussions, we believe that the MTEA can successfully capture the magnitudes and spatio-temporal variations of PPM and SPM in China. However, there are still some uncertainties in the model estimation and its application in China.

Firstly, the assumption of non-significant correlation between PPM versus SPM may be violated by the fact that $SO_2$ and $NO_x$ emitted from combustions will further generate secondary sulfate and nitrate particulates. Nevertheless, the combustion processes for generating $SO_2/NO_x$ and PPM are still different. PPM, i.e. BC and POC, mainly comes from incomplete combustion of residential activities, such as burning biofuels and coal (Long et al., 2013), but $SO_2$ and $NO_x$ mainly come from the complete combustion process of industrial and transportation sources, such as coal, gasoline and diesel (Lu et al., 2011; Li et al., 2017b;

Tang et al., 2019). In addition, the MTEA approach uses the assumption of non-significant correlation rather than irrelevance. Such processing also reduces uncertainty to a certain extent.

Secondly, natural sources of PPM, such as fine dust from desert and sea salt, are not taken into account in the MTEA approach. As a result, PPM in the city near a desert or sea could be underestimated. For example, the $PM_{2.5}$ components observational campaign conducted in 2015 showed that the contribution of sea salt aerosols to ambient $PM_{2.5}$ mass concentration in Haikou is 3.6-8.3% (Liu et al., 2017).

Thirdly, current bottom-up emission inventories are generally outdated with a time lag of at least 1-2 years, mainly due to the lack of timely and accurate statistics. Consequently, the adjoint uncertainty in MTEA estimation is inevitable. To evaluate the uncertainty, a comparison test was conducted by adjusting the apportioning coefficient (the a and b in Eq. 1) with a disturbance of ±0.1. Firstly, we decreased the value of a in each populous city by 0.1. Meanwhile, the coefficient b increased by 0.1. This scenario indicates an overestimation in contribution of combustion-related process to primary $PM_{2.5}$ or underestimation in contribution of dust-related process. Secondly, we increased the value of a in each populous city by 0.1 (decreased b by 0.1) for checking the opposite case. The results are presented in Table S5 and point out that the estimated secondary proportions of $PM_{2.5}$ varied less than ±3% in most populous cities caused by the changes of the apportioning coefficient. This sensitivity experiment highlights that the apportioning coefficients depending on emissions has limited impacts on the final estimation results. Generally, the uncertainty of apportioning coefficient is one of two factors that directly affect the tracer X. The other one is the concentration of CO and PMC itself. Hence, we also conducted a similar test to check the impacts of tracer X on the model estimation by changing the tracer concentrations mentioned in Eq.1. Specifically, we (1) increased CO concentration by 10% as well as decreased PMC concentration by 10% and (2) decreased CO concentration by 10% as well as increased PMC concentration by 10%. Both sets of adjustment show changes within ±2% in the estimated secondary proportions of $PM_{2.5}$ in all cities except for Urumqi (Table S6). This phenomenon from the perspective of tracer concentration also supports that the impacts of the tracer X on the final model results are limited. In summary, we believe that the most determinative stuff for the final results of our model is the principle of the minimum correlation between PPM and SPM but not the tracer X which relies on emissions or concentrations.

**5 Conclusions**

In this study, we developed a new approach MTEA to distinguish the primary and secondary compositions of $PM_{2.5}$ efficiently from routine observation of $PM_{2.5}$ concentration with much less computation cost than traditional CTMs. By comparing with long-term and short-term measurements of aerosol chemical components in China as well as aerosol composition network in the United States, we showed that MTEA was able to capture variations of PPM and SPM concentrations. Meanwhile, our model posed a great agreement with the reanalysis dataset from one of the most advanced CTMs in China as well.

The method was then applied to the surface air pollutant concentrations from MEE observation network in China, and offered an effective way to understand the characteristics of PPM and SPM covering a wide area. In terms of spatial pattern, MTEA reveals that SPM accounts for 63.5% of total $PM_{2.5}$ in southern cities averaged for 2014-2018, while in the North the proportion drops to 57.1%. It should be noted that the secondary proportion in regional background regions is ~19% higher than that in populous regions. In terms of seasonality, the estimated national averaged secondary proportion is the lowest in fall (56.1%), and for the other three seasons it stays among 61%.

Moreover, we applied MTEA to explore the changes of secondary proportion in $PM_{2.5}$ in China. In recent years, the $PM_{2.5}$ pollution in China has been significantly alleviated benefiting from a series of emission control measures. The MTEA results suggest that both PPM and SPM are decreased simultaneously in populous regions, while for regional background regions, the reduction of secondary $PM_{2.5}$ is much more notable than the PPM. The secondary proportion of $PM_{2.5}$ in populous cities during 2014-2018 keeps constant (56.4-58.5%) in general on an annual average scale, but it poses a slight but consistent increase in summer, mostly due to the elevated $O_3$ and stronger photochemistry pollution in China. In addition, with the help of MTEA, we found that the secondary $PM_{2.5}$ proportion in Beijing significantly increased by 34% during the COVID-19 lockdown, which might be the main reason for the observed unexpected PM pollution in this special period.

Finally, we applied MTEA to explore the synergistic correlation between $PM_{2.5}$ and $O_3$. Estimated results demonstrate that PPM is weakly correlated with $O_3$, its effect on $O_3$ is mainly to inhibit the production of $O_3$ via adjusting radiation balance and affecting radical level. While SPM is positive correlated with $O_3$ in presence of the effect of AOC. Higher ambient $O_3$ concentration indicates stronger AOC, and further leads to more SPM generation.

We suggested that the regional differences in the correlation between total $PM_{2.5}$ and $O_3$ are mainly caused by the different PPM levels in BTH and YRD regions.

We also discussed the uncertainties of MTEA method. MTEA may pose overestimation on the secondary fractions of $PM_{2.5}$ in those regions which are near to desert or sea by ~20% for failing taking natural dust into consideration. In addition, the sensitivity experiment through imposing reasonable disturbance on emissions and tracer concentrations also show the limited impacts on final estimation. Overall, the most determinative stuff for our model estimate is the principle of the minimum correlation between PPM and SPM.

China has been plagued by $PM_{2.5}$ pollution in recent years. Different $PM_{2.5}$ compositions may have varying impacts on environment, climate and health, due to the different sources and generation pathways. Therefore, it's of great importance to quantify PPM and SPM for the pollution recognition and prevention. The methods to quantify different $PM_{2.5}$ components are often based on either lab analysis of offline filter samplings or online observation instruments such as AMS. However, these methods are often labor-intensive, strict technical and high economic cost. CTM is another useful tool to reveal the composition characteristics of $PM_{2.5}$. But traditional CTMs are short in high requirement of hardware as well. Our study develops an efficient approach based on statistical principle to explore PPM and SPM with lower cost, and applying this approach to large-scale observation networks, such as the MEE network, can offer an unprecedented opportunity to quantify the $PM_{2.5}$ components on a large space and time scale.

**Code and Data availability.** The MTEA software package and input datasets are available at http://nuistairquality.com/m_tea. Observational datasets and modeling results in the text are available upon request to the corresponding author (linan@nuist.edu.cn).

**Author contribution.** NL designed this study. NL and HL supervised this work. HRZ and KQT established, performed and improved MTEA model. HRZ and NL interpreted the data and wrote the original draft. CH, HLW, SG and MH provided the long-term measurements of aerosol compositions. HL, CS, JLH, XLG, MDC, ZXL and HY provided useful comments on the paper, and all authors contributed to the revision of the manuscript.

**Competing interests.** The authors declare that they have no conflict of interest.

**Acknowledgements.** This work was supported by the National Key Research and Development Program of China (2018YFC0213802 and 2019YFA0606804), the National Natural Science Foundation of China (41975171), and the Major Research Plan of the National Social Science Foundation (18ZDA052). The numerical calculations in this paper have been done on the supercomputing system in the Supercomputing Center of Nanjing University of Information Science & Technology.

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

**Table 1.** Seasonal mean concentrations of the primary and secondary $PM_{2.5}$ in 31 populous cities and 19 regional background cities of China.

| City | PPM ($\mu g \cdot m^{-3}$) | | | | SPM ($\mu g \cdot m^{-3}$) | | | | SPM/PM$_{2.5}$ (%) | | | |
|---|---|---|---|---|---|---|---|---|---|---|---|---|
| | MAM | JJA | SON | DJF | MAM | JJA | SON | DJF | MAM | JJA | SON | DJF |
| *Populous cities in the Northern China* | | | | | | | | | | | | |
| Beijing | 31.0 | 28.4 | 30.6 | 34.1 | 25.0 | 23.7 | 20.1 | 16.2 | 44.7 | 45.4 | 39.6 | 32.2 |
| Tianjin | 17.8 | 13.7 | 21.9 | 28.2 | 42.0 | 35.3 | 32.9 | 29.0 | 70.2 | 72.1 | 60.0 | 50.7 |
| Shijiazhuang | 35.0 | 22.4 | 41.5 | 54.0 | 36.7 | 35.5 | 32.1 | 37.7 | 51.2 | 61.3 | 43.6 | 41.1 |
| Taiyuan | 22.0 | 20.2 | 32.7 | 32.3 | 28.4 | 22.2 | 21.0 | 25.0 | 56.3 | 52.3 | 39.1 | 43.6 |
| Hohhot | 13.1 | 11.4 | 18.2 | 20.1 | 19.2 | 13.1 | 16.0 | 20.7 | 59.5 | 53.6 | 46.8 | 50.7 |
| Shenyang | 21.0 | 16.7 | 24.4 | 27.8 | 26.1 | 17.4 | 20.8 | 28.0 | 55.3 | 51.0 | 46.0 | 50.2 |
| Changchun | 21.3 | 15.8 | 20.2 | 28.9 | 18.3 | 12.3 | 17.2 | 25.0 | 46.2 | 43.9 | 46.0 | 46.4 |
| Harbin | 14.1 | 9.3 | 15.5 | 27.2 | 25.5 | 15.2 | 20.9 | 38.9 | 64.4 | 61.9 | 57.3 | 58.9 |
| Jinan | 25.6 | 23.0 | 29.9 | 32.4 | 38.2 | 30.7 | 30.7 | 38.3 | 59.9 | 57.1 | 50.7 | 54.2 |
| Zhengzhou | 24.8 | 20.2 | 28.6 | 34.1 | 45.2 | 28.8 | 33.9 | 44.1 | 64.6 | 58.7 | 54.3 | 56.4 |
| Lhasa | 6.6 | 5.9 | 8.2 | 5.8 | 13.0 | 9.2 | 9.3 | 13.6 | 66.3 | 61.2 | 53.2 | 70.1 |
| Xian | 24.1 | 15.3 | 31.3 | 37.1 | 31.5 | 20.1 | 24.5 | 41.3 | 56.7 | 56.7 | 44.0 | 52.7 |
| Lanzhou | 14.1 | 10.1 | 17.8 | 21.3 | 29.3 | 24.1 | 24.8 | 33.2 | 67.6 | 70.4 | 58.2 | 60.9 |
| Xining | 14.8 | 12.4 | 18.3 | 17.9 | 26.4 | 19.3 | 21.0 | 34.5 | 64.1 | 60.8 | 53.4 | 65.9 |
| Yinchuan | 12.9 | 8.2 | 16.1 | 18.7 | 22.8 | 21.8 | 21.1 | 27.0 | 63.8 | 72.8 | 56.7 | 59.1 |
| Urumqi | 15.2 | 9.5 | 16.5 | 27.9 | 30.9 | 19.1 | 32.0 | 63.6 | 67.1 | 66.9 | 66.0 | 69.5 |
| **Average** | 19.6 | 15.2 | 23.2 | 28.0 | 28.7 | 21.7 | 23.6 | 32.3 | 59.4 | 58.9 | 50.4 | 53.5 |
| *Regional background cities in the Northern China* | | | | | | | | | | | | |
| Weihai | 8.1 | 7.1 | 8.6 | 10.7 | 23.8 | 18.5 | 14.9 | 13.7 | 74.6 | 72.2 | 63.4 | 56.0 |
| Jiayuguan | 7.8 | 7.0 | 7.5 | 7.0 | 16.6 | 11.4 | 14.5 | 19.2 | 68.1 | 61.9 | 65.9 | 73.4 |
| Zhangjiakou | 10.8 | 11.0 | 10.7 | 10.7 | 14.2 | 14.4 | 12.8 | 14.4 | 56.8 | 56.6 | 54.5 | 57.4 |
| Daxinganling | 4.3 | 3.6 | 4.6 | 5.7 | 9.2 | 7.7 | 9.3 | 11.6 | 68.0 | 67.9 | 67.0 | 66.9 |
| Xilingol | 2.3 | 2.3 | 2.8 | 3.1 | 10.2 | 9.3 | 7.7 | 9.1 | 81.8 | 80.1 | 73.1 | 74.7 |
| Yanbian | 9.9 | 5.6 | 9.4 | 11.7 | 15.3 | 9.1 | 13.5 | 17.4 | 60.7 | 62.1 | 58.9 | 59.7 |
| Guyuan | 12.3 | 9.0 | 11.9 | 13.1 | 19.0 | 13.1 | 14.7 | 20.1 | 60.7 | 59.2 | 55.4 | 60.6 |
| Yushu | 4.3 | 2.1 | 4.2 | 3.9 | 10.0 | 9.6 | 7.1 | 9.9 | 69.8 | 82.3 | 62.7 | 71.5 |
| Altay | 2.0 | 1.3 | 1.7 | 2.7 | 6.3 | 6.3 | 6.0 | 8.0 | 76.1 | 83.5 | 77.5 | 74.7 |
| **Average** | 6.9 | 5.5 | 6.8 | 7.6 | 13.8 | 11.1 | 11.2 | 13.7 | 66.9 | 67.0 | 62.1 | 64.2 |
| *Populous cities in the Southern China* | | | | | | | | | | | | |
| Shanghai | 12.4 | 11.1 | 11.7 | 15.8 | 29.5 | 22.5 | 20.8 | 25.4 | 70.4 | 67.0 | 64.1 | 61.6 |
| Nanjing | 19.1 | 16.0 | 19.9 | 24.3 | 29.2 | 18.7 | 19.9 | 28.5 | 60.4 | 53.9 | 50.1 | 54.0 |
| Hangzhou | 21.1 | 17.8 | 21.5 | 23.6 | 24.9 | 14.5 | 18.9 | 28.5 | 54.1 | 45.0 | 46.8 | 54.7 |
| Hefei | 16.4 | 14.6 | 17.9 | 23.2 | 39.8 | 26.7 | 30.1 | 39.8 | 70.9 | 64.6 | 62.7 | 63.2 |
| Fuzhou | 9.0 | 7.5 | 7.5 | 7.6 | 18.0 | 12.9 | 13.7 | 19.7 | 66.6 | 63.3 | 64.7 | 72.2 |
| Nanchang | 14.8 | 9.8 | 13.2 | 15.8 | 20.6 | 13.6 | 22.3 | 28.8 | 58.2 | 58.1 | 62.9 | 64.6 |
| Wuhan | 18.5 | 15.6 | 18.9 | 25.3 | 36.4 | 19.9 | 30.0 | 45.3 | 66.3 | 56.1 | 61.3 | 64.2 |
| Changsha | 17.6 | 13.2 | 17.5 | 21.9 | 31.5 | 21.1 | 31.2 | 40.0 | 64.1 | 61.5 | 64.1 | 64.6 |
| Guangzhou | 11.6 | 9.5 | 12.1 | 12.7 | 22.6 | 16.3 | 23.4 | 26.6 | 66.0 | 63.3 | 65.9 | 67.7 |
| Nanning | 11.7 | 9.7 | 14.9 | 13.3 | 22.0 | 12.9 | 19.9 | 28.7 | 65.3 | 57.1 | 57.1 | 68.3 |
| Haikou | 5.8 | 4.7 | 8.1 | 6.0 | 11.5 | 6.9 | 8.7 | 15.8 | 66.3 | 59.4 | 51.8 | 72.6 |
| Chongqing | 17.9 | 14.0 | 18.6 | 21.6 | 24.1 | 19.4 | 25.0 | 38.8 | 57.5 | 58.0 | 57.3 | 64.2 |
| Chengdu | 29.6 | 20.0 | 27.1 | 31.7 | 23.6 | 15.0 | 18.2 | 39.1 | 44.3 | 42.8 | 40.1 | 55.2 |
| Guiyang | 13.5 | 10.6 | 12.2 | 9.9 | 21.3 | 12.2 | 18.5 | 29.8 | 61.2 | 53.6 | 60.4 | 75.0 |
| Kunming | 9.3 | 6.5 | 6.9 | 8.1 | 21.1 | 13.5 | 16.1 | 18.4 | 69.5 | 67.6 | 69.9 | 69.3 |

| Average | 15.2 | 12.0 | 15.2 | 17.4 | 25.1 | 16.4 | 21.1 | 30.2 | 62.2 | 57.7 | 58.1 | 63.5 |
|---|---|---|---|---|---|---|---|---|---|---|---|---|

### *Regional background cities in the Southern China*

| | | | | | | | | | | | | |
|---|---|---|---|---|---|---|---|---|---|---|---|---|
| Huangshan | 5.3 | 5.1 | 5.7 | 6.4 | 20.7 | 11.2 | 16.3 | 22.7 | 79.5 | 68.8 | 74.2 | 78.1 |
| Nanping | 6.1 | 5.0 | 6.4 | 5.7 | 15.9 | 11.4 | 13.4 | 17.4 | 72.2 | 69.7 | 67.9 | 75.4 |
| Zhoushan | 9.5 | 8.0 | 8.4 | 11.9 | 13.7 | 10.2 | 10.1 | 11.5 | 59.2 | 56.2 | 54.5 | 49.1 |
| Shanwei | 7.9 | 4.8 | 8.2 | 5.7 | 16.6 | 10.3 | 17.4 | 22.7 | 67.8 | 68.2 | 68.1 | 79.9 |
| Beihai | 7.5 | 4.2 | 10.6 | 8.7 | 16.4 | 8.2 | 16.4 | 25.8 | 68.7 | 65.9 | 60.6 | 74.7 |
| Qianxinan | 3.3 | 1.7 | 2.2 | 2.9 | 12.5 | 12.1 | 12.2 | 13.8 | 79.2 | 87.9 | 84.8 | 82.9 |
| Sanya | 4.6 | 4.2 | 5.5 | 3.7 | 9.7 | 5.6 | 6.8 | 11.7 | 67.8 | 56.8 | 55.4 | 75.8 |
| Aba | 2.0 | 2.1 | 2.1 | 2.9 | 10.5 | 10.3 | 10.3 | 10.8 | 84.2 | 83.0 | 83.2 | 78.7 |
| Linzhi | 2.3 | 1.5 | 2.0 | 2.1 | 7.5 | 6.2 | 5.3 | 7.6 | 76.6 | 80.5 | 73.0 | 78.5 |
| Diqing | 1.9 | 1.5 | 1.7 | 1.6 | 10.5 | 9.4 | 9.4 | 10.2 | 84.7 | 86.4 | 84.8 | 86.2 |
| **Average** | 5.0 | 3.8 | 5.3 | 5.2 | 13.4 | 9.5 | 11.7 | 15.4 | 72.7 | 71.4 | 69.1 | 74.9 |

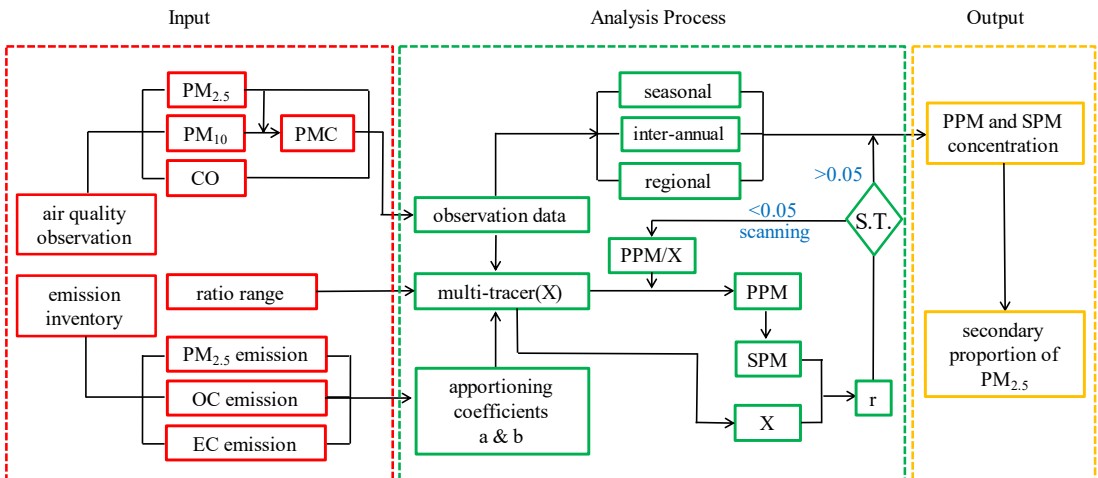

**Figure 1**. The flow chart of the M-TEA approach. The part in red indicates the air quality data and emission input. The part in green stands for the key process for predicting PPM/SPM based on the routine PM$_{2.5}$ observation. In this part, S.T. means the significant test. The significant level α is set to 0.05. The part in orange indicates the final output.

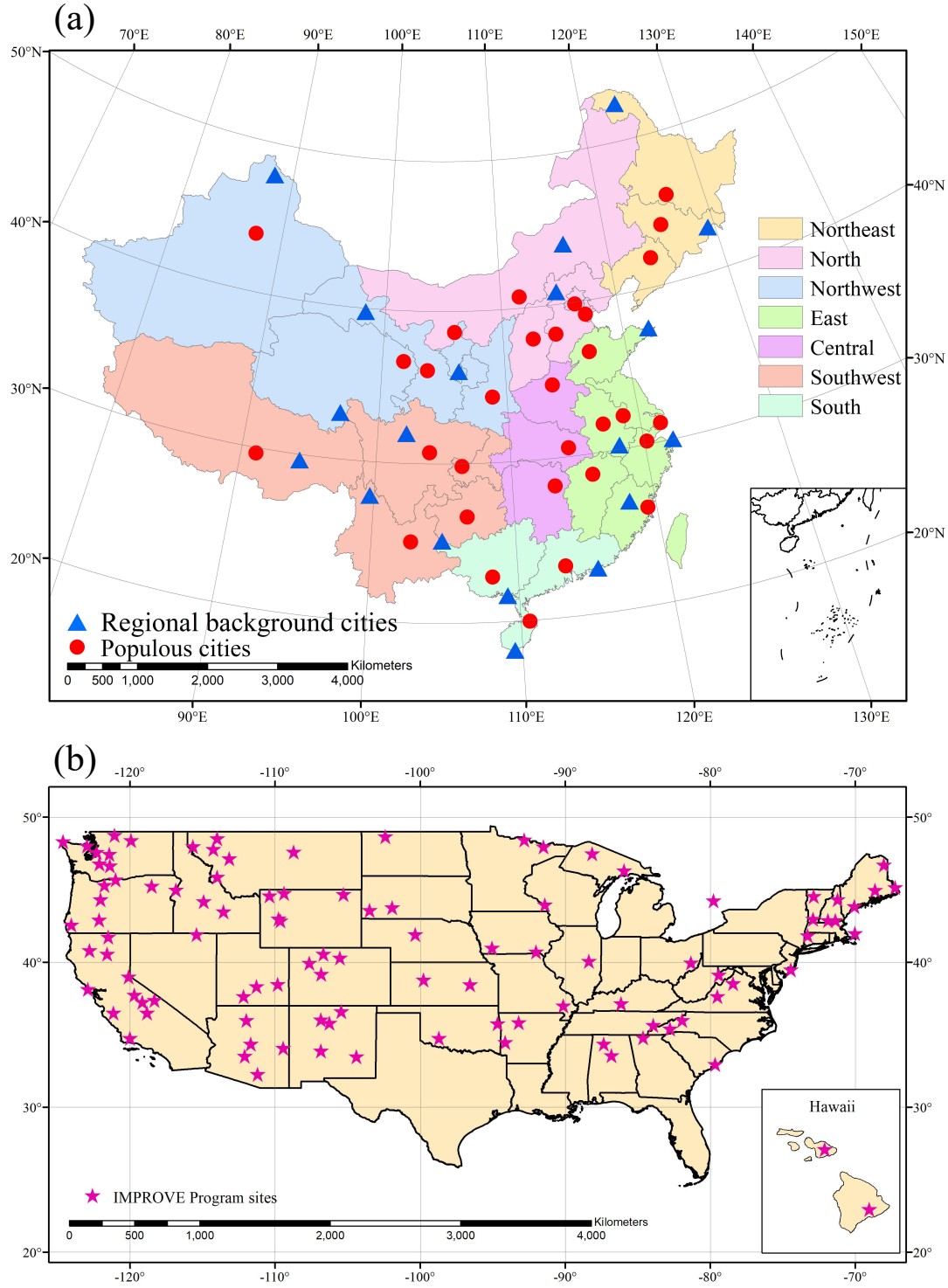

**Figure 2.** The geographical locations for the observational data used in this study. (a) Geographical locations of 31 populous cities (red circles) and 19 regional background cities (blue triangles) of China in this study. (b) Spatial distribution of the IMPROVE aerosol monitoring network (pink pentagrams) in the United States.

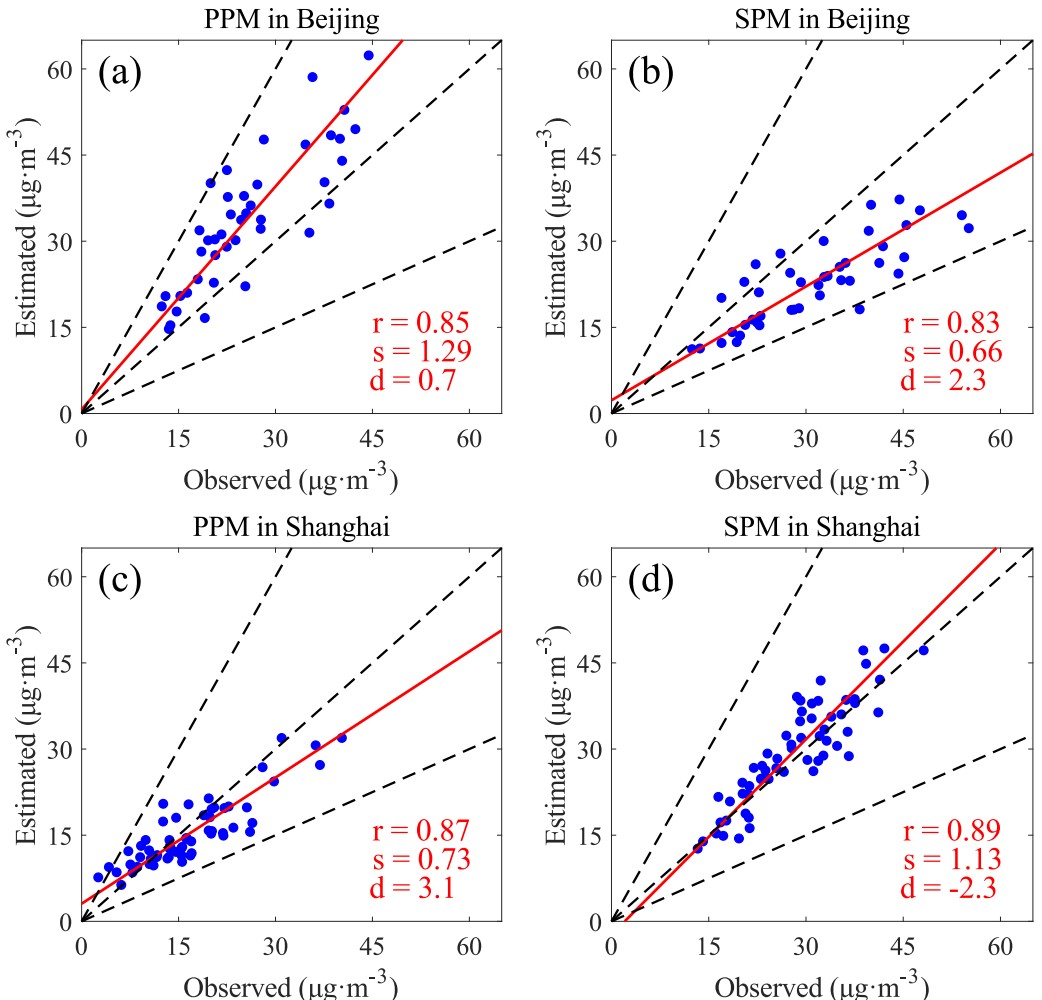

**Figure 3.** The scatter evaluation between the monthly mean of observed PM versus that of estimated PM in Beijing (a-b) and Shanghai (c-d), China. Panel (a, d), (b, e) denotes PPM and SPM. The red numbers in each panel indicate the Pearson correlation coefficient (r), the slope (s) and the intercept of fitting line (d). The fitting lines in red were based on the Reduced Major Axis (RMA) regression. The black dotted line in each panel from left to right represents 2:1, 1:1 and 1:2 ratio respectively.

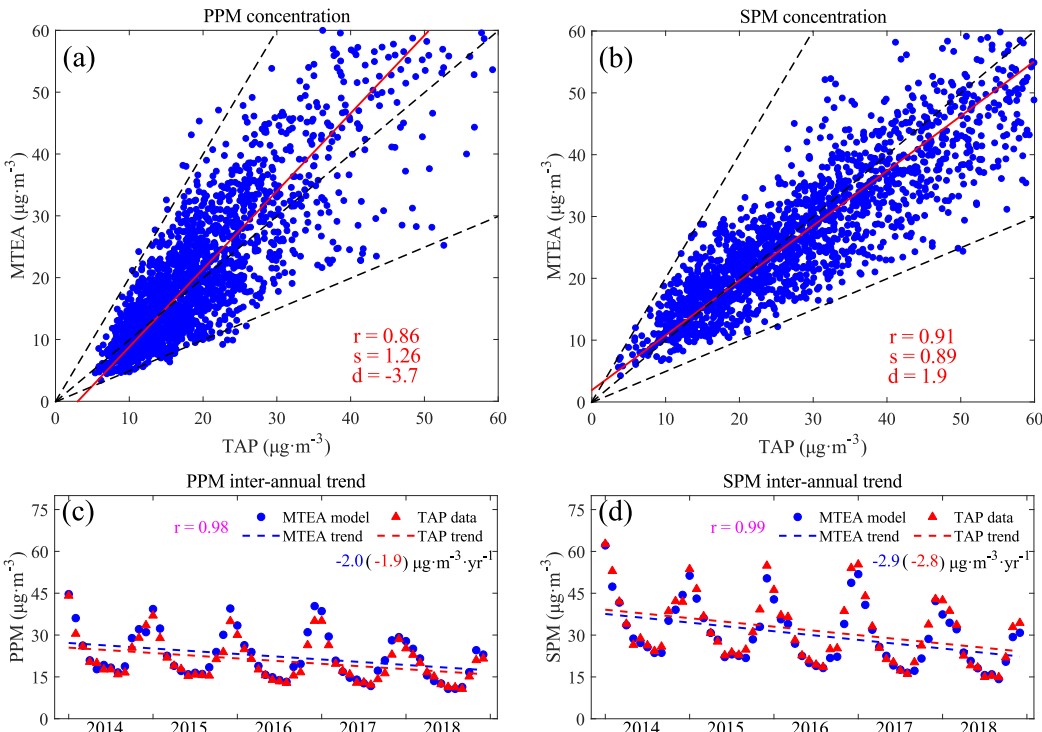

**Figure 4.** Comparisons between MTEA and TAP in terms of PPM, SPM concentrations and their annual trends from 2014 to 2018 in 31 populous cities of China. In panel (a) and (b), each blue solid dot stands for a monthly mean concentration of PPM or SPM in one of 31 populous cities. The number of samples is 1860 (60×31). The metrics r, s and d represent correlation coefficient, slope and intercept of fitting line, respectively. The fitting method follows the Reduced Major Axis (RMA) regression. In panel (c) and (d), MTEA and TAP are marked by blue circles and red triangles. Each dot represents the mean PPM/SPM concentration of 31 cities. The colorful numbers stand for the annual trends of PPM and SPM concentrations during 2014-2018. At the same time, the correlation coefficient (r) between MTEA versus TAP is also provided.

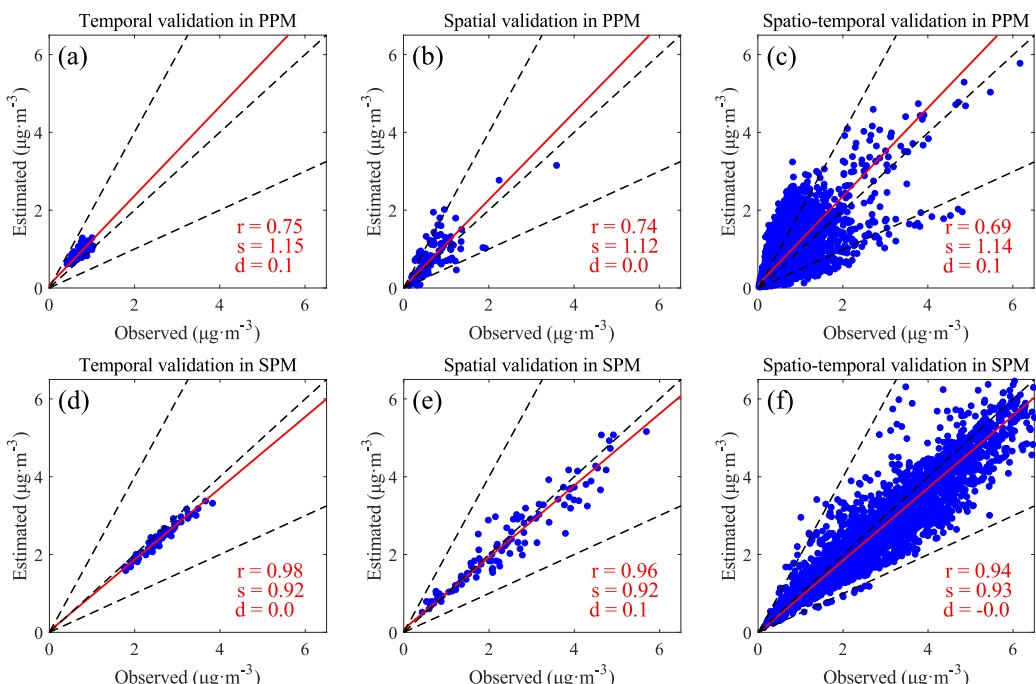

**Figure 5.** The scatter evaluation between the monthly mean of observed PPM(a-c)/SPM(d-f) versus that of estimated PPM/SPM in the United States. Panel (a, d), (b, e) and (c, f) denotes temporal, spatial and spatio-temporal mixed validation respectively. The red numbers in each panel indicate the Pearson correlation coefficient (r), the slope (s) and the intercept of fitting line (d). The fitting lines in red were based on the RMA regression. The black dotted line in each panel from left to right represents 2:1, 1:1 and 1:2 ratio respectively.

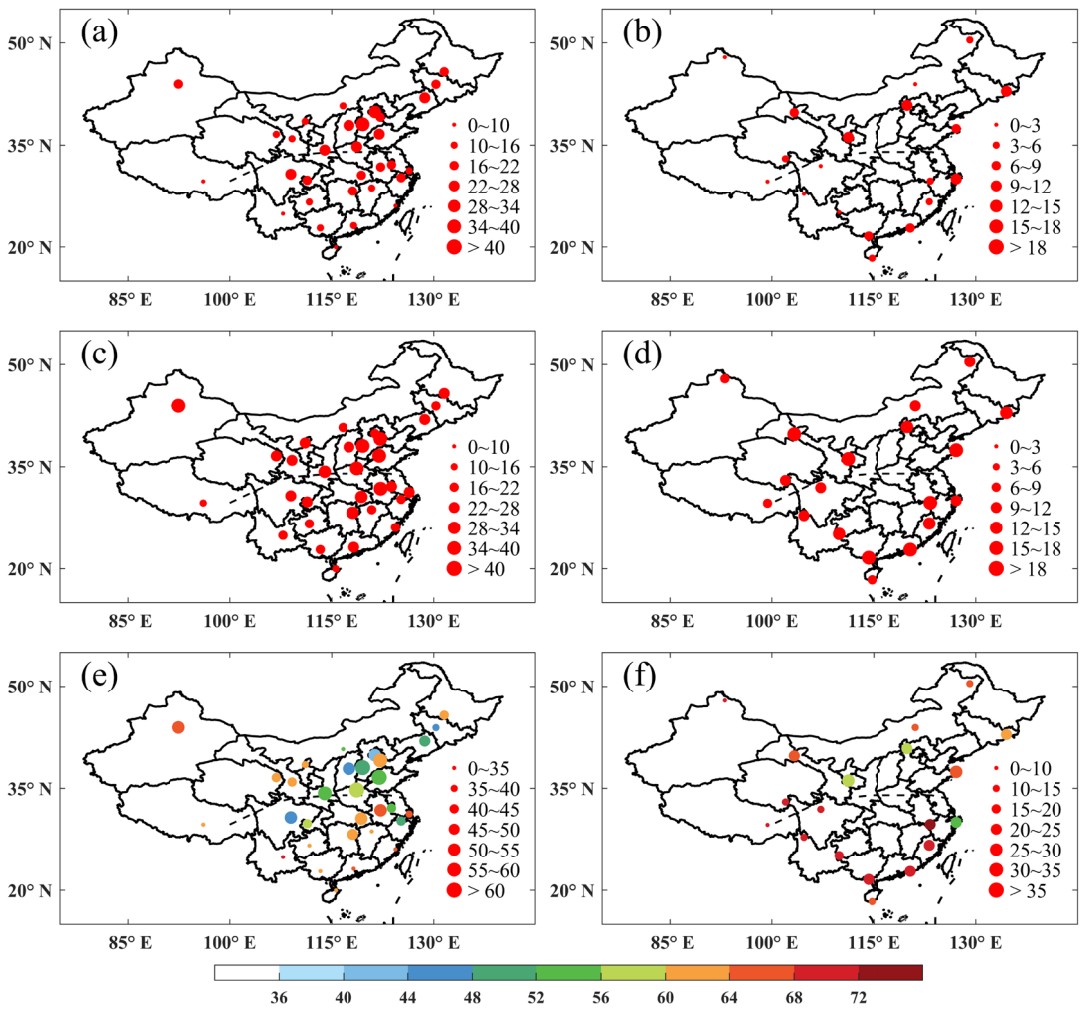

**Figure 6.** Spatial distributions of PPM (a, b), SPM (c, d) and total PM₂.₅ concentration (e, f) averaged for the studying period. The secondary proportions of PM₂.₅ (SPM/total PM₂.₅) are also shown in (e, f). The left column (a, c, e) indicates populous cities. The right column (b, d, f) is for the regional background cities. The black dotted line in each panel shows the Qinling-Huaihe line.

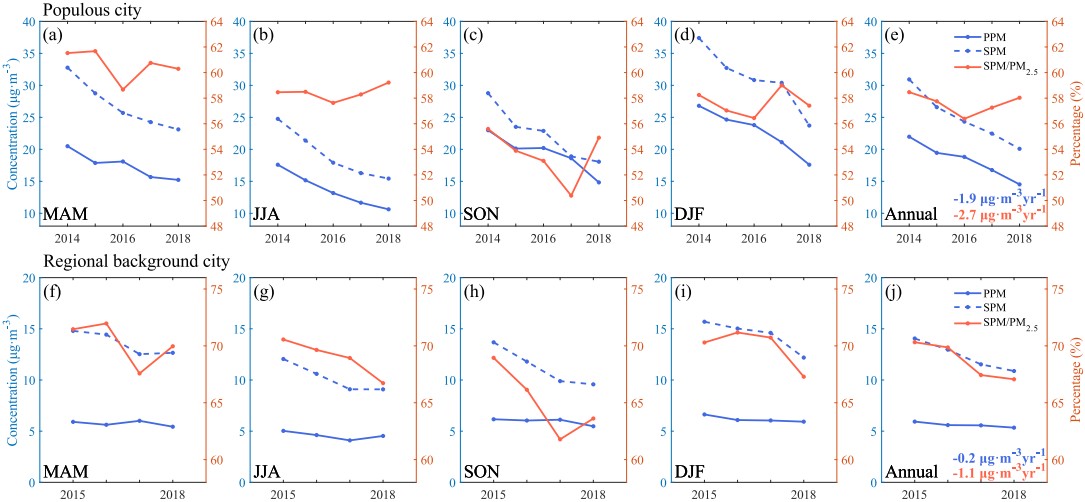

**Figure 7.** Inter-annual variations of PPM concentrations (blue solid line), SPM concentrations (blue dotted line) and the secondary proportions of PM$_{2.5}$ (red solid line) in populous cities (a-e) and regional background cities (f-j). MAM (a, f), JJA (b, g), SON (c, h) and DJF (d, i) denotes spring, summer, fall and winter respectively. The absolute decreases in PPM/SPM concentration are labeled in blue/red near the panel (e, j).

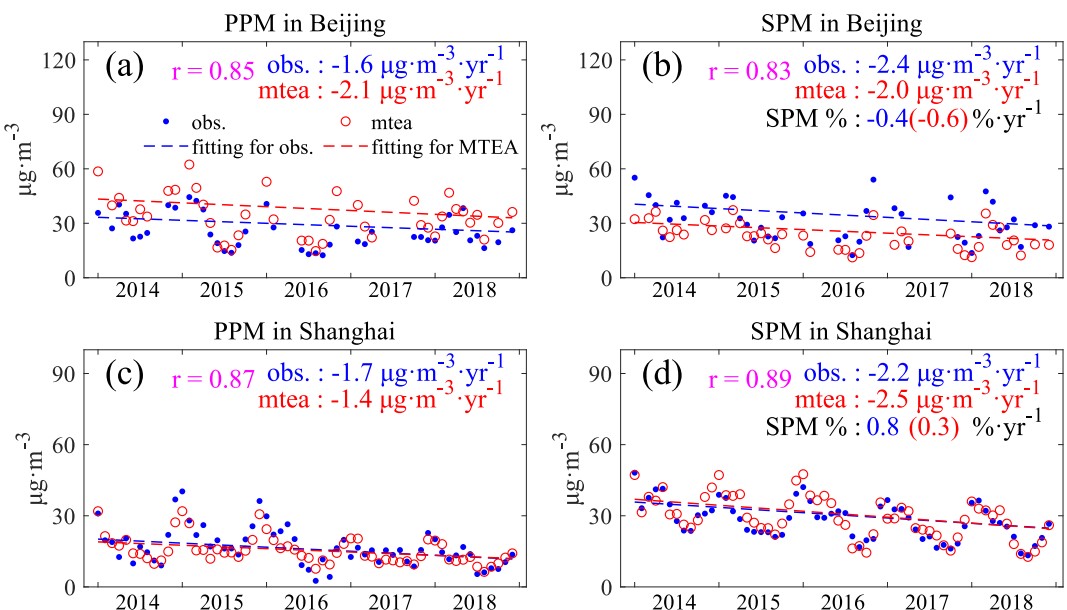

**Figure 8.** The monthly time series variation of PM in Beijing (a-b) and Shanghai (c-d). Panel (a, d), (b, e) denotes PPM, SPM respectively. In each panel, in-situ observation and MTEA estimation is shown in blue and red dots. Meanwhile, bule and red dotted line stands for the long-term trend in concentration changes. The values of the decrease rates in PPM and SPM concentrations as well as the relative changes in the secondary proportions of PM$_{2.5}$ (SPM %) are also provided at the upper right corner of each panel.

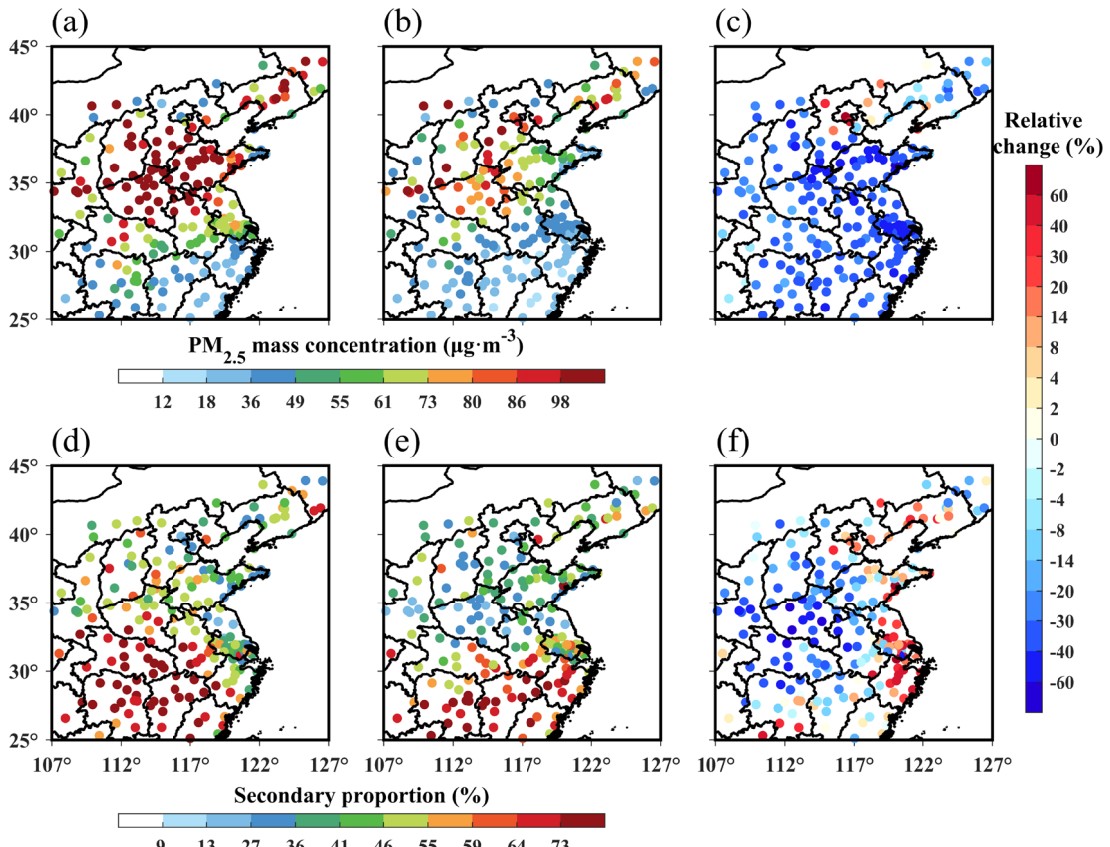

**Figure 9.** The application of M-TEA in estimating PPM/SPM during the COVID-19 lockdown. Panel a and b denotes the spatial distribution of PM$_{2.5}$ mass concentration before the national lockdown (01~23 Jan 2020, pre-lockdown) and during the national lockdown (23-Jan ~ 17-Feb 2020, post-lockdown). And panel c indicates the relative change between panel a and panel b, i.e. (post-lockdown – pre-lockdown)/pre-lockdown. Panel (d-f) is the same as panel (a-c), but for the secondary proportions of PM$_{2.5}$.

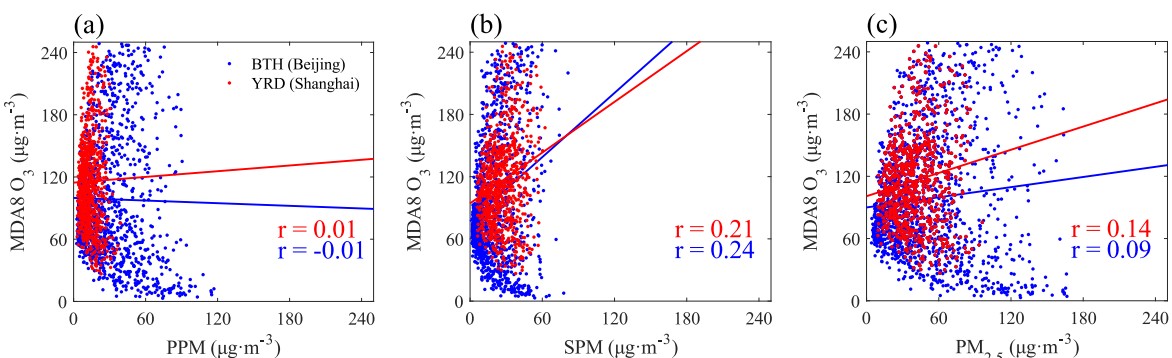

**Figure 10.** Scatter plot about the correlation between daily PM concentration and MDA8 $O_3$ concentration in Beijing (blue) and Shanghai (red). Based on the reanalysis dataset ERA5 from ECMWF, those days when precipitation took place were removed. Panel a-c indicates PPM, SPM and total $PM_{2.5}$ respectively. In each panel, solid-colored lines represent the fitting line based on Least Squares method. The Peason correlation coefficient (r) are also given at the bottom right of the panels.