# Peer review of "Estimation of Secondary PM2.5 in China and the United States using a Multi-Tracer Approach"

_Atmospheric Chemistry and Physics, 2021_

## Referee Comment (RC2)

The manuscript demonstrates the multi-tracer estimation algorithm (MTEA), to identify the primary and secondary components from routine observation of PM$_{2.5}$, and validates the method by comparing the long-term and short-term measurements of aerosol chemical composition in China and a network from the United States. This method provides a useful and uncomplicated way to estimate primary and secondary PM, using routine observation species and emission inventories. This manuscript aims to address important questions quantifying primary and secondary aerosols and is within the scope of ACP.

However, regarding the method itself, the method should be carefully introduced with more details. The validation part is a bit weak, and should be strengthened in the next version. It is vital because only with good validation can one trust the result from the model. In addition, in the result and discussion part, the discussion is superficial, which needs to be improved in depth, and backed up by more scientifc evidence and/or publications.

As a conclusion, the manuscript provides a novel algoriths in primary and secondary particle concentrations, however, the manuscript is not carefully written from the perspective of science and scientific writting, with certain degree of improvement for publication in ACP. Therefore, this manuscript needs a major revision in terms of major context and English language.

**Major comments:**

- Introduction: the introduction is poorly written and need to be re-write. If I were you, I would write the introduction based on this outline: 1) introduction of atmospheric aerosols, including sources, type, chemical composition and impacts on air quality, human health and climate, 2) summarise other studies, you must state what has been achieved and what is the current challenging, 3) what is your paper about, how this paper can narrow the gap.

  In the current version, the point 1) is addressed, but should be introduced in smoother way. The author is trying to address the point 2), but the studies mentioned in the paragraph 3 in page 3 look not very relavent. For example, the author summarises the online and offline studies, which is good, and people can see the drawbacks of field and lab measurement to study the PPM and SPM, so the next paragraph should state to overcome these drawbacks, people use model to study the PPM and SPM, and should also state what these model studies have achieved and/or the drawbacks of these method. Finally, this paragraph can lead the final paragraph in the introduction, namely, introduce this study and how this study advances the model studies on PPM and SPM estimation.

- Methodology: the methodology part is written in a reasoble logic, but the author needs to pay more attention to speficy the technical details, e.g., the definition of some terms.
- Model validation: this part straightforwardly deliver the good validation result between model and observation. Good correlation is shown in this part, suggesting good model performance. However, this part also requires more interpretation on the model's over/underestimation behaviour compared to observation, which is now absent. Ideally, the author should foucs most on this part, because only when the model is reasonable validated can we trust the result and make the further interpration on the result. Therefore, from my own perspective, the author should strengthen this part.
- Result and discussion: this part also very straighforwardly and logically reports the results. However, the interpretation of results should be more comprehensive and backed up by previous studies and/or solid evidence, which is absent now and needs to be added. In addition, the discussion of the result is very superficial, lacking depths, which should also be improved.
- Conclusion: it summarises the significance of the study, but one or two paragraph need to be re-written, based on the revised context in Section 4.

**Other comments:**

**Introduction**:

-Page 3 Line 4: please check the size of these emission sources. Dust resuspension and brake wear are typically in coarse mode. Also sea salt and sea spray by definition have certain degree of overlap.

-Page 3 Line 6: please specify how the secondary aerosol is formed, e.g., gas-particle partitioning mechanism for new particle formation.

-Page 3 paragraph 1: please restructure this paragraph, by introducing primary and secondary, instead of saying "one source….and the other source….". Better still, avoid to say the sources of $PM_{2.5}$, because the sources you listed as primary sources produce more coarse mode particles, rather than accumulation mode and/or Aitken mode, e.g., dust, sea spray.

-Page 3 Line 18: please give an example of the toxicities on human health.

-Page 3 Line 21: please define "individual" and "short-term", and make it more clear.

-Page 3 Line 26: please give a range of percentage to support the word "dominant".

-Page 3 Line 28: please also mention the instrument and method using in this study.

-Page 3 Line 29: actually, this sentence describes the general situation. Even if you don't cite this study, you still can make the statement. Please specify what the significance of the study and how it support the main point in this paragraph.

-Page 3 Line 31: please note that the offline filter measurement is tyically considered as a cheaper method compared to the online measurement. Using offline filter measurement, we can collect samples from various sampling sites for a long period.

-Page 4 Line 5: to make it consistent, it should be "Section 3" instead of "section 3".

-Page 4 Line 14: MEE is not defined in the previous text.

**Methodology**:

-Page 4 Line 24: why specifically CO and why don't select black carbon? Here the author states that he/she uses the PMC to track flying dust, but in the introduction, the author states dust belongs to the size range of $PM_{2.5}$. Please specify.

-Page 4 Line 31: "fine mode" is not defined. It seems that it refers to $PM_{2.5}$, but please specify and give the definition of "fine mode" and "coarse mode" in the study.

-Page 5 Line 3: "OC" is not defined in the previous text.

-Page 5 Line 14: good to show the table, but it is better to include 0.01 and 0.05 case, as a sensitivity test.

-Page 5 Line 25: why is this range and why is this range reasonable?

-Page 5 Line 26: "each varying ratio may obtain a series of SPM, along with a coefficient of"? What does the "series" mean? Do you mean time series?

-Page 5 Line 27: what is the R (coefficient of determination)? Is it a Pearson correlation coefficient? How is it defined? How is it calculated? Please also introduce Figure S1 and explain it a bit here or in the SI.

-Page 5 Line 27: please also introduce Figure S1 and explain it a bit here or in the SI. Regarding the y-axis, does it refer to $R^2$ or $p$-value? What does the $p$-value mean?

-Page 5 Line 27: how do you know if primary and secondary PM are from the same or different sources? What if they are from similar sources, which side of the green are would you choose?

-Page 6 Line 13: MEE is not defined in the previous text. Which region does MEE network cover, whole China?

-Page 6 Line 16: under what guideline and/or operation standard does MEE do the measurement? Presumably, Ambient air quality standards (GB 3095-2012)?

-Page 6 Line 20: in Table S3, is the $PM_{2.5}$ concentration the annual mean $PM_{2.5}$ concentration? Please speficy.

-Page 6 Line 24: "the major gaseous and particle pollutants" sounds awkward. Maybe change it to "the major gaseous pollutants and particle". Here particle pollution is presumable PM, right? You can just say "the major gaseous pollutants and particulate matter".

-Page 6 Line 26: just saying "for exmaple" is fine.

-Page 6 Line 30: awkward here. Please read the sentence again, it is hard to identify the verb in the sentence. Does the author want to say something like: these studies employ offline filter-based measurement? If yes, please change the word "employed" to "employ".

-Page 7 Line 1: change "compositions" to "composition", the same applies to Page 7 line 8

-Page 7 Line 1: change "for directly comparing" to "to directly compare".

-Page 7 Line 16: still, please speficy the region covered by this network.

-Page 7 Line 24: why does the author lower the time resolution? Presumably for the computational efficiency? If so, please specify.

**Model validation**:

-Page 8 Line 23: why is the RMA regression model chosen? Please specify the reason. Has the author tried other regression models?

-Page 8 Line 24: why are the values over "30 μg·m$^{-3}$ higher than the city average" removed? Please specify the reason. In addition, why is the number 30 μg·m$^{-3}$ chosen?

-Page 8 Line 7: in the figure, the author specifies the "r", "s" and "d" in the caption, but not here. Please also explain the meaning here.

-Page 8 Line 9: what does "NMB" mean? How is it calculated?

-Page 8 Line 9: please also try to interpret the reason of over/underestimation.

-Page 8 Line 13: why not report "NMB" here?

-Page 9 Line 6: here the author is trying to interpret the discrepencies between model and observation. However the sentence here looks like he/she is trying to summarise the general reasons for discrepencies by saying "However, we find that there are still a few discrepancies between the estimated and observation-based results, and the main reasons might be", but the author specifies the reason for for discprencies for some individual sites. Therefore, the author can state that the he/she observed the discrepencies in some cities, and then specify the reasons for each site.

-Page 9 Line 19: how are the spatial, temporal and spatio-temporal correlation calculated? I guess for spatial, the author gets the average concentration of each site and plot them in the estimation vs obversation space? Please speficy how they are calculated.

-Page 9 Line 20: awkward, there is one value for estimated concentration and one value for observed concentration, then the author plots them in the estimation vs obversation space. So for each pair, there is only one point for one time point or one site.

-Page 9 Line 26: here, please also speficy why the Pearson correlation coefficient for SPM is higher than it for PPM.

-Page 9 Line 29: here, please also speficy why the slope for PPM is higher than it for SPM.

**Result and discussion**:

-Page 10 Line 11: In Figure 5, what is the size of the solid circle? The legend is missing here for this information. Please specify this.

-Page 10 Line 18: the high energy consumption cannot directly link to the serious pollution, it also depends on what sort of energy. In the northern part of China, power plants combust coal for electricity and in the winter for heating as well. But we use nuclear power plants for this purpose, we don't have air pollution at all even if we have high energy comsumption. This is the point. However, in the south, heating provided by the government and power plants is not very common. That is reason why China has this contrast between the north and the south. Please check the literature to back up this statement.

-Page 10 Line 24: please see how the atmospheric condistion in the south is more favourable for SPM, and back up this argument with scientific literature.

 -Page 10 Line 31: primary emission can also be regionally transported, This interpretation is not strong.

-Page 11 Line 9: the boundary layer height and temperature inversion is the major reason that particle concentration is higher at night than during the day. The author can simply say the high level of pollution is related to high emission from biomass buring and coal combustion as it is in the paper, and the more stagnent air condition (which means bad mixing condition) in the winter than summer.

-Page 11 Line 13: replace the "is" after PPM and SPM by "of"

-Page 11 Line 16: please interpret why the contribution is lower in autumn, and what this phenomena suggests.

-Page 11 Line 20: please explain why AOC is higher in warm season, and back this up by literature.

-Page 11 Line 22: this is a weak argument, because the primary pollutant can also be transported from the north to the south. The SPM can also be formed during the transport, or the air mass passes through a region which can facilitate the SPM formation. So there are many possibilities. In addition, please also add plots indicating the trajectories of air masses from the north to the sourth, or add some literature to support your statement of monsoon transport.

-Page 11 Line 27: it is an issue of English language. It should be "dotted blue line" and "solid blue line".

-Page 12 Line 4: it is another issue of English language. The author uses past tense in Section 3, but the present tense here. Please use consistent tense throughout the paper.

-Page 12 Line 6: please cite studies to justify the interpretation here about ozone pollution.

-Page 12 Line 11: not convinced by the interpretation, please provide more evidence from your analysis and/or other studies.

-Page 12 Line 17: in the figure caption of Figure 7. Line 3 in the caption, "bule" should be "blue". Please check the grammar and conjugate the verbs correctly for plural and third person singular.

-Page 13 Line 3: "were decreased" should be "decreased", and also change for "increase" likewise.

-Page 13 Line 5: please define "BTH" in the previous text, or when it first appears in the main text.

-Page 13 Line 7: "its averaged value" here is a bit nebulous. Please specify the length of the period being averaged before COVID.

-Page 13 Line 8: here the present tense is used, whereas in two lines, a past tense verb is used. Please describe your result in a consistent tense.

-Page 13 Line 11: in the Figure 8 caption, "The application of M-TEA in estimating PPM/SPM during the COVID-19 lockdown" here has an typo, and it should be "The application of MTEA in PPM/SPM estimation during the COVID-19 lockdown". The word "post" in the phrase "post-lockdown" in line 4 means "after", but it is not what the author means here.

-Page 13 Line 17: I guess the author would like to say the condition is not good for atmospheric mixing. Please check the difference between mixing and diffusion.

-Page 13 Line 18: "was increased" should be "increased". Please check this throughout the text.

-Page 13 Line 21: "YRD" is not defined in the previous text. Please define it.

-Page 13 Line 26: not accurate. The $PM_{2.5}$ is the particulate matter with aerodynamic diamter smaller than 2.5 μm, which can be solid particle or liquid droplet. The $PM_{2.5}$ can be directly emitted from primary sources, and some species in $PM_{2.5}$ can be formed via chemical reactions. If the particles are directly emitted, then the word "precursor" in the sentence is wrong. Ozone in the troposphere is mainly produced by the reactions of $NO_x$ and VOCs.

-Page 13 Line 32: please check the Figure S3 caption. There is typo "bule", which should be "blue"

-Page 14 Line 1: please also include the cities which fail in the test in the figure when you reply the referee's comment.

-Page 14 Line 2: please conjugate the verb correctly. "positive relationship" should be presumbly "positive correlation"?

-Page 14 Line 3: the correlation coefficient "r" here should be *italic* "*r*". Please change it throughout the text.

-Page 14 Line 9: please cite these "series of recent studies".

-Page 14 Line 13: "compare" should be "compared".

-Page 14 Line 14: the definition of BTH and YRD should be addressed in the early text.

-Page 14 Line 20: Figure 9 caption has some typos and mistakes in grammar. Which method is used specifically for "Least Squares method"? The trust-region Levenberg-Marquardt least orthogonal distance regression?

-Page 14 Line 20: This paragraph should be rewritten. It is hard to accept that the correlation coefficient *r* smaller than 0.3 shows some meaningful correlation, espeically those with *r* about 0.09 and 0.14. There are very closed to 0, which means not correlated at all. From the subfigures in Figure 9, it is also hard to say there is any correlation.

-Page 15 Line 12: Should "reduces" be "increases"?

**Conclusion**:

-Page 16 Line 2: Should "offered" be "provided"?

-Page 16 Line 20: Please re-think about this paragraph.

-Page 16 Line 27: This paragraph looks like a repetition of the introduction. It should foucs more on the outlook, e.g., how your study advance the current understanding, and what implication it might provide for future studies.

---

## Author Comment (AC1)

**Response to RC#1:**

Dear Editor and anonymous referee #3:

We greatly appreciate your consideration and the reviewer's constructive comments on the manuscript of "Estimation of Secondary PM$_{2.5}$ in China and the United States using a Multi-Tracer Approach" (acp-2021-683). We have carefully revised the manuscript to address all the comments as described below. Reviewer comments are shown in blue. Our responses are shown in black. The revised texts are shown in italics.

This study developed a new method to determine the portion of primary and secondary PM$_{2.5}$ using some basic measurements and inventory. They evaluated this new approach through the comparison with lots of observations in China and US. In addition, they analyzed the temporal and spatial variation as well as correlation between O$_3$ and PM$_{2.5}$ using the results from their new method. Although their evaluation looks very well, I think their results were not enough convincing because of unclear statement of their method and defect of this method. I would suggest major revision before reconsideration. My detail comments are following.

**Response:** We thank the reviewer for the comments. According to the reviewer's helpful and insightful comments, we have revised our manuscript and the point-by-point responses to the specific comments were given subsequently. We sincerely hope the revisions are able to address the reviewer's concerns.

1. Eq (1) and Eq (2): These equations are the core of their method. They regarded CO as one tracer to represent the combustion process and assumed the combustion emission sources are same for CO, OC and EC. This assumption is mostly correct, but the emission factor/emission ratio of CO, OC and EC from different combustion sources are different. I think it is unconvincing to use one single coefficient without the influence of diversity of sources to standard for all conditions. I may misunderstand something, please discuss this uncertainty or make this clear.

**Response:** Thanks for the conducive comments. We also do believe that the emission factors of CO, OC and EC from different sources are various as well. Our method tracks the combustion process, which produces OC and EC, by regarding CO

as the tracer. However, the correlation between different sources of diverse carbonaceous matter is hard to find out with the aid of current routine observations of

CO. The coefficients in Eq. 1 are aimed at representing the relative contribution of combustion process and flying dust to primary $PM_{2.5}$. We constrained the uncertainty of both coefficients by setting up a set of sensitivity tests. The specific discussion about this uncertainty is in Section 4.5. The specific configuration issue your concerned about the sensitivity experiment will be clarified in the following 3rd point.

The final experiment result indicates that the adjustment of coefficient for CO (*a*)

within 0.1 does not obviously affect the estimated secondary proportions of $PM_{2.5}$ (<

3%). To make this point clearer, the detailed description of this part has been corrected in the revised manuscript as follows.

***Revision in Section 2.1:***

*As shown in Eq. 1, we use a and b to quantify the relative contributions of*

*combustion and dust process to PPM. Given that the complicated process such as the*

*combustion from multiple sources is hard to represent via current routine CO*

*observations, we avoid considering the correlation among these sources but focus on*

*the relative weights of combustion process and flying dust. Meanwhile, the*

*uncertainty resulting from the apportioning coefficient a and b will be further*

*discussed in Section 4.5.*

2. Eq (2): why did you name b as emission of fine dust? To my knowledge,

MEIC does not include the emission of dust even urban dust.

**Response:** Thanks for your concerns. The dust emissions are not specifically separated from $PM_{2.5}$ emissions in MEIC. In fact, the composition of $PM_{2.5}$ emission in MEIC includes EC, OM, sulfate, nitrate and other trace elements such as Al, Ca, Si, Fe, Mg, K and other species etc. (Li et al., 2017a). Trace elements are usually related to the flying dust from constructions and onroad traffic transportation. In the MTEA approach, we would like to represent the dust-related part of PPM with the emissions of the mineral dust in fine mode particulate matter. We calculated the dust-related emissions by deducting the emissions of EC, OM, sulfate and nitrate from total $PM_{2.5}$ emissions. We revised the relevant texts for a clearer statement.

***Revision in Section 2.1:***

*Coefficient b is aimed at reflecting the activity intensity of fine mode dust by counting its emissions. However, MEIC does not directly provide fine mode dust emissions. It is included in the emissions of total $PM_{2.5}$ (Li et al., 2017a). Thus we inferred the fine mode dust emission by deducting the emissions of EC, POA, sulfate and nitrate from the $PM_{2.5}$ emissions.*

3. I did not understand how you did the sensitivity experiment to examine the uncertainty in the inventories. Page 16, you said you changed the emission coefficient with 10%. If so, how can you keep a+b=100%? According to my understanding on this new method, the results should have large dependence on the inventory of $PM_{2.5}$, OC, EC even the factor you used to decide OA, $SO_4^{2-}$ and $NO_3^-$. I would strongly suggest setting up more comprehensive and scientific sensitivity experiments to discuss the dependence on the inventory.

**Response:** Thank you for your conducive comments and rigorous attitude to scientific research. Coefficients *a* and *b* are determined by calculating the relative ratio between EC+POA to dust as Eq. 1-2. Hence the uncertainty of emission inventory can lead to the changes of the ratio *a* to *b*. In Section 4.5, we tested the adjoint changes of the final estimated secondary proportions of $PM_{2.5}$ by adjusting the coefficient *a*. The adjustive test includes two parts. Firstly, we increased the value of *a*

in each city by 0.1 to check the model results in the case of underestimating the contributions of combustion process (or overestimating the contributions of dust process). Under this circumstance, the coefficient $b$ which represents dust process should be decreased by 0.1. On the contrary, we also decreased the value of $a$ in each city by 0.1 to check the model results in the case of overestimating the contributions of combustion process (or underestimating the contributions of dust process).

Meanwhile, the coefficient $b$ which stands for dust process is increased by 0.1. The sum of $a$ and $b$ is still 100%. The sensitivity experiment results indicate that the disturbance of coefficient $a$ ($\pm$0.1) lead to changes in the secondary proportions of

$PM_{2.5}$ within $\pm$3% (refer to Table S5 in the supplementary material). In addition, the discussion about the uncertainty of the primary sulfate and nitrate emissions also reveals that the predicted results are not sensitive to their emissions (refer to Section

2.1 and Table S1 in the supplementary material). Therefore, we indeed agree that the emission inventory can pose impacts on our model estimation, but the effects are not obvious.

The assumed tracer of PPM (i.e. X, see Eq. 1) is one of the cores of MTEA

approach. However, the most determinative stuff for the final results of our model is the principle of the minimum correlation between PPM and SPM but not only the value of the tracer X. To prove this view, we also carried out another kind of test in adjusting X by changing the concentrations of CO and PMC. We (1) increased CO

concentration by 10% as well as decreased PMC concentration by 10% and (2)

decreased CO concentration by 10% as well as increased PMC concentration by 10%.

Both sets of adjustment demonstrate changes within $\pm$2% in the estimated secondary proportions of $PM_{2.5}$ in all cities except for Urumqi (Table R1). This phenomenon also supports that the impacts of the tracer X on the final model results are not obvious. To clearly state the point mentioned by the reviewer, we have rephrased the relevant texts in the manuscript.

***Revision in Section 2.1:***

*We evaluated the potential effect of the coefficient, by conducting a set of*

*comparative simulation with the coefficient of 0.03, and found that the final estimated*

*SPM was not sensitive to this coefficient (Table S1). Thus we concluded that the*

*uncertainty of primary sulfate and nitrate emissions did not significantly influence the*

*final estimation of MTEA model. For other uncertainties of X which are dependent on*

*emission intensities or tracer concentrations, we would conduct discussions in the*

*later Section 4.5.*

***Revision in Section 4.5:***

*To evaluate the uncertainty, a comparison test was conducted by adjusting the*

*apportioning coefficient (the a and b in Eq. 1) with a disturbance of ±0.1. Firstly, we*

*decreased the value of a in each populous city by 0.1. Meanwhile, the coefficient b*

*increased by 0.1. This scenario indicates an overestimation in contribution of*

*combustion-related process to primary $PM_{2.5}$ or underestimation in contribution of*

*dust-related process. Secondly, we increased the value of a in each populous city by*

*0.1 (decreased b by 0.1) for checking the opposite case. The results are presented in*

*Table S5 and point out that the estimated secondary proportions of $PM_{2.5}$ varied less*

*than ±3% in most populous cities caused by the changes of the apportioning*

*coefficient. This sensitivity experiment highlights that the apportioning coefficients*

*depending on emissions has limited impacts on the final estimation results. Generally,*

*the uncertainty of apportioning coefficient is one of two factors that directly affect the*

*tracer X. The other one is the concentration of CO and PMC itself. Hence, we also*

*conducted a similar test to check the impacts of tracer X on the model estimation by*

*changing the tracer concentrations mentioned in Eq.1. Specifically, we (1) increased*

*CO concentration by 10% as well as decreased PMC concentration by 10% and (2)*

*decreased CO concentration by 10% as well as increased PMC concentration by 10%.*

*Both sets of adjustment show changes within ±2% in the estimated secondary*

*proportions of $PM_{2.5}$ in all cities except for Urumqi (Table S6). This phenomenon from*

*the perspective of tracer concentration also supports that the impacts of the tracer X*

*on the final model results are limited. In summary, we believe that the most*

*determinative stuff for the final results of our model is the principle of the minimum*

*correlation between PPM and SPM but not the tracer X which relies on emissions or*

*concentrations.*

**Table R1.** Impacts of tracer concentration uncertainty on the estimated secondary proportion of

$PM_{2.5}$ [1] in China (Unit: %).

| City | Secondary proportion of $PM_{2.5}$ | Change of secondary proportion of $PM_{2.5}$ | |
|---|---|---|---|
| | | 1.1 * CO concentration & 0.9 * PMC concentration | 0.9 * CO concentration & 1.1 * PMC concentration |
| Beijing | 40.3 | -0.01 | 0.01 |
| Tianjin | 61.9 | -0.32 | -0.52 |
| Shijiazhuang | 44.8 | -0.26 | -0.28 |
| Taiyuan | 43.1 | 0.22 | 0.17 |
| Hohhot | 48.6 | -0.03 | -0.01 |
| Shenyang | 48.7 | -0.06 | -0.06 |
| Changchun | 47.9 | 0.03 | 0.04 |
| Harbin | 66.9 | 0.22 | -0.59 |
| Shanghai | 68.0 | -1.51 | -1.90 |
| Nanjing | 50.3 | 0.00 | 0.03 |
| Hangzhou | 45.6 | -0.42 | -0.46 |
| Hefei | 65.4 | -1.57 | -1.73 |
| Fuzhou | 64.8 | -0.25 | -0.44 |
| Nanchang | 62.5 | -0.33 | -0.42 |
| Ji'nan | 54.6 | -0.04 | -0.02 |
| Zhengzhou | 54.6 | 0.14 | 0.14 |
| Wuhan | 61.5 | -1.45 | -1.49 |
| Changsha | 65.9 | -1.60 | -1.74 |
| Guangzhou | 65.2 | 0.00 | -0.28 |
| Nanning | 65.2 | -0.22 | -0.47 |
| Haikou | 65.9 | -0.15 | -0.09 |
| Chongqing | 62.7 | -0.23 | -0.31 |
| Chengdu | 45.3 | 0.42 | 0.44 |
| Guiyang | 65.6 | -0.22 | -0.50 |
| Kunming | 70.4 | -0.40 | -0.69 |
| Lhasa | 56.1 | 0.07 | 0.05 |
| Xi'an | 52.6 | -0.04 | -0.01 |
| Lanzhou | 60.0 | 0.15 | 0.02 |
| Xining | 59.1 | -0.56 | -0.60 |

| | | | |
|---|---|---|---|
| Yinchuan | 59.5 | 0.02 | -0.06 |
| Urumqi | 72.1 | -2.70 | -2.85 |

[1] Based on the MEE observations in 2016.

4. Figure 3, as I saw, the largest concentration is < 60 μg/m3. Why not short the range of axis to spread those dots?

**Response:** Thanks for your highly careful reminding. We have reduced the range of axis from 130 to 65 for aesthetics. And the revised figure is shown below.

***Revision in Fig. 3:***

[Figure]

5. P8L7: Why did you remove the heavy pollution cases here as well as in Section 4? As you stated at P10L25, you would like to avoid the influence of extreme high primary emission cases. However, mostly heavy pollution cases are caused by unfavored meteorological condition but not caused by sudden high primary emission (except the biomass burning cases). I would be curious that how your method applied to analyze the heavy pollution cases. In general, it is more important to understand the contribution of secondary particles to heavy pollution cases than the general conditions.

**Response:** Thanks for your highly conducive comments and rigorous attitude to
scientific research. The data preprocessing in P8L7 and P10L25 are different. The
data preprocessing in Section 3.1.1 is aimed at removing the gap between long-term
measurements of $PM_{2.5}$ at a single site and routine observation of $PM_{2.5}$ from national
network for further evaluation.

However, the data preprocessing in Section 4 is prepared for the usage of data
from MEE. To address reviewer's concern, we take estimation in 2016 as an example
and make a comparison. MTEA method shows that the estimated secondary
proportions of $PM_{2.5}$ without excluding the heavy polluted cases are 2.0-13.7% lower
than that including the data preprocessing (Fig. R1). We agree with the reviewer that
unfavorable meteorological conditions are major causes for haze events. Under these
unfavored meteorological conditions, the assumed tracer X may have extremely high
co-linear relationship with total $PM_{2.5}$. Thus the PPM concentrations may be falsely
overestimated. Here we excluded these days to avoid the incorrectly estimation and
focus more attention on the common characteristics of PPM/SPM during the general
periods. We revised the statement in Section 3.1.1 and Section 4 for a clearer version.

*Revision in Section 3.1.1:*

*Given the discrepancy in $PM_{2.5}$ concentrations between in-situ measurements of*
*a single site and multiple MEE national sites, we firstly preprocessed the data for*
*further evaluation. In data preprocessing, we removed the in-situ daily measurements*
*whose value was over 30 $\mu g \cdot m^{-3}$ higher than the city average (from MEE).*

*Revision in Section 4:*

*The observations during severe haze events (top 10% CO and PMC polluted*
*days) were excluded to avoid the influence of unfavorable meteorological conditions*
*and extreme high primary emission cases. Unfavorable meteorological conditions are*
*major causes for haze events. PPM under these unfavored meteorological conditions*
*may have considerable high co-linear relationship with total $PM_{2.5}$. The concentration*

*of SPM from complicated formation pathways is then underestimated. Therefore, we*

*excluded these polluted days to focus more attention on general characteristics of*

*PPM and SPM concentration.*

[Figure]

**Figure R1.** The estimated secondary proportions of PM$_{2.5}$ in case of including (No_Ex_top_10%)

and excluding top 10% polluted days (Ex_top_10%) in 2016.

6. P10L30: Could you explain what is regional background cities you defined here? Usually, cities are not background.

**Response:** Thank you for pointing this out. We agree that cities usually are not categorized as background regions. We are aimed at disclosing the discrepancy in

PPM/SPM among diverse cities which depend on different levels of anthropogenic activity. The 19 regional background cities in this study are chosen because they suffered the least PM$_{2.5}$ pollution during 2014-2018. The averaged mean PM$_{2.5}$

concentration of each city is less than 35.0 μg m$^{-3}$ (National Ambient Air Quality

Standard level II of China, NAAQS) except for Guyuan, Ningxia Province (refer to

Table S3 in the supplementary material). We believe that these selected cities can generally reveal the PM pollution characteristics of the regions which are under sparse anthropogenic emissions. For a clearer expression, we have revised the related texts in the manuscript.

***Revision in Section 2.2.1:*** *31 among the 50 cities are provincial capital cities,*

*employed to represent populous cities, while the rest 19 relatively small cities are*

*categorized as regional background cities (Table S3). The mean PM$_{2.5}$ concentration*

*of each regional background city is less than 35 μg m⁻³ (National Ambient Air Quality Standard level II of China, NAAQS) except for Guyuan, indicating that they are slightly impacted by anthropogenic activities. By comparing populous cities with regional background cities, we could reveal the discrepancy in PPM and SPM among those regions which suffer from different levels of PM$_{2.5}$ pollution.*

7. Section 4.2.1: I think the seasonal variation of PPM and SPM is largely depend on the seasonal variation of emissions you applied.

**Response:** Thank you for your comments. We indeed agree with the reviewer that the seasonal pattern of the estimated PPM and SPM concentration can be attributed to the seasonal variations of emissions. Taking Shanghai as an example, we tested the impacts of the seasonal variations of emissions on the estimated PPM and SPM concentrations by comparing two cases (i.e. seasonal emissions in this study and homogenous emissions in the ideal sensitivity experiment). As listed in Table R2, though the seasonal maxima/minima of PPM and SPM concentration still happen in the wintertime/summertime, but the specific concentrations significantly change. The maximum of relative change can be 10% (PPM in DJF, changes from 15.8 μg·m⁻³ to 14.3 μg·m⁻³).

**Table R2.** Comparison of seasonal PPM and SPM concentrations between applying seasonal emissions or homogenous emissions in Shanghai (Unit: μg·m⁻³).

|  |  | MAM | JJA | SON | DJF |
|---|---|---|---|---|---|
| PPM | Seasonal emissions (This study) | 12.4 | 11.1 | 11.7 | 15.8 |
|  | Homogenous emissions (Ideal study) | 12.8 | 11.7 | 12.2 | 14.3 |
| SPM | Seasonal emissions (This study) | 29.5 | 22.5 | 20.8 | 25.4 |
|  | Homogenous emissions (Ideal study) | 29.2 | 21.9 | 20.3 | 26.8 |

 8. Section 4.2.2: Did you use the emission inventory for specific year here?
China conducted a large reduction on PM$_{2.5}$ emission since 2014. If you did not use
the specific inventory, the estimated trend of PPM and SPM would not make sense,
even though they agreed with observations. In addition, could you show the
correlation coefficient between the observation and estimation here?

**Response:** Thanks for your concern. We indeed agree with the reviewer's
opinion that the emission inventory should be matched for each year. For
anthropogenic emissions from 2014 to 2017, we utilized the MEIC emission
inventory (v1.3) developed by Tsinghua University, which is publicly offered at their
website (http://meicmodel.org/) (Li et al., 2017a; Li et al., 2017b). In terms of
emissions after 2017, we also accessed from MEIC support team (Zheng et al., 2021).
For the correlation coefficient between the observation and estimation in Section 4.2.2,
we have followed the suggestion from the reviewer and showed it both in the related
figure and the related texts in the manuscript.

[Figure]

***Revision in Section 4.2.2:***

*Applying the MTEA model to this case, we are delighted to find that our model*
*not only successfully reproduces the consistent decreasing trends of PPM and SPM in*
*Beijing and Shanghai (correlation coefficient r of observation versus estimation*
*ranges from 0.83 to 0.89), but also captures the different trends in secondary*

*proportions of $PM_{2.5}$ in the two cities (–0.6% $yr^{-1}$ in Beijing and 0.3% $yr^{-1}$ in*

*Shanghai).*

9. Section 4.3: The same issue as above. Did you update the inventory to the lockdown condition? If yes, please state the inventory you used here and the decrease in the emission of $PM_{2.5}$, CO, OC, EC.

**Response:** Thank you for your concern. We used the emission reduction ratio in of various air pollutants during the COVID-19 lockdown from Huang et al. (2020). The specific emission reduction ratios of various air pollutants are listed in Table R3. Meanwhile, we modified the related texts in the manuscript to make it clearer.

***Revision in Section 4.3:*** *With the help of MTEA, we tracked variations of the secondary proportions of $PM_{2.5}$ in East China before and during the COVID-19 lockdown (Fig. 8 d-f). The specific emission reductions owing to the national lockdown were derived from Huang et al. (2020). Based on the bottom-up dynamic estimation, provincial emissions of CO, $NO_x$, $SO_2$, VOC, $PM_{2.5}$, BC and OC decreased by 13-41%, 29-57%, 15-42%, 28-46%, 9-34%, 13-54%, and 3-42%, respectively during the lockdown period.*

**Table R3.** Estimation of provincial emission reduction ratio (%) of CO, $NO_x$, $SO_2$, VOC, $PM_{2.5}$, BC, OC due to COVID-19 lockdown in China.

| Province | CO | $NO_x$ | $SO_2$ | VOC | $PM_{2.5}$ | BC | OC |
|---|---|---|---|---|---|---|---|
| Beijing | 22% | 45% | 26% | 45% | 18% | 46% | 8% |
| Tianjin | 21% | 38% | 20% | 41% | 14% | 22% | 6% |
| Hebei | 15% | 45% | 16% | 36% | 12% | 17% | 5% |
| Shanxi | 18% | 40% | 20% | 33% | 16% | 19% | 10% |
| Inner Mongolia | 14% | 29% | 15% | 34% | 13% | 16% | 6% |
| Liaoning | 21% | 40% | 28% | 36% | 16% | 28% | 8% |
| Jilin | 16% | 39% | 23% | 34% | 13% | 18% | 5% |
| Heilongjiang | 17% | 37% | 27% | 28% | 13% | 15% | 7% |
| Shanghai | 35% | 48% | 42% | 45% | 34% | 54% | 42% |
| Jiangsu | 23% | 50% | 26% | 41% | 16% | 35% | 7% |

| | | | | | | | |
|---|---|---|---|---|---|---|---|
| Zhejiang | 41% | 50% | 29% | 45% | 30% | 49% | 20% |
| Anhui | 14% | 56% | 22% | 31% | 11% | 22% | 4% |
| Fujian | 29% | 51% | 30% | 42% | 19% | 31% | 7% |
| Jiangxi | 24% | 53% | 21% | 43% | 19% | 30% | 9% |
| Shandong | 23% | 50% | 25% | 39% | 19% | 35% | 9% |
| Henan | 23% | 57% | 22% | 41% | 18% | 35% | 8% |
| Hubei | 19% | 55% | 23% | 35% | 16% | 23% | 10% |
| Hunan | 22% | 51% | 25% | 36% | 20% | 24% | 15% |
| Guangdong | 38% | 50% | 33% | 46% | 27% | 42% | 13% |
| Guangxi | 24% | 50% | 28% | 39% | 17% | 27% | 5% |
| Hainan | 24% | 44% | 25% | 36% | 14% | 25% | 4% |
| Chongqing | 18% | 53% | 32% | 37% | 14% | 20% | 4% |
| Sichuan | 16% | 50% | 27% | 33% | 9% | 15% | 3% |
| Guizhou | 24% | 39% | 25% | 30% | 22% | 25% | 20% |
| Yunnan | 24% | 51% | 25% | 41% | 18% | 21% | 8% |
| Tibet | 16% | 35% | 15% | 35% | 14% | 14% | 5% |
| Shaanxi | 19% | 45% | 18% | 34% | 13% | 22% | 5% |
| Gansu | 13% | 47% | 16% | 29% | 9% | 13% | 3% |
| Qinghai | 23% | 46% | 22% | 39% | 20% | 20% | 7% |
| Ningxia | 24% | 36% | 24% | 39% | 20% | 23% | 8% |
| Xinjiang | 16% | 35% | 15% | 35% | 14% | 14% | 5% |

10. Section 4.4: How did you decide the diurnal variation of emission? Was your result sensitive to the diurnal pattern? Because the diurnal pattern of $O_3$ concentration is almost constant.

**Response:** Thank you for your careful concerns. MEIC provides the bottom-up anthropogenic emission inventory with monthly time resolution. Based on the fixed total emissions, we further distributed them with specific diurnal variation patterns of each sector, including power plants, industry, transportation and residential sources (Fig. R3a). This kind of preprocessing is also adopted for preparing emission input for other air quality model studies and is proved to be reasonable (Li et al., 2021; Zhang et al., 2021).

We used the processed emissions as input for MTEA method and found that the model results show obvious diurnal pattern as well. The diurnal patterns are characterized by two peaks in the day, one occurring at 10:00-15:00 (local time,

UTC+8) and the other appearing at 22:00-00:00. The 10:00-15:00 peak can be explained by the elevating emissions of $PM_{2.5}$ precursors, such as $NO_x$ and $SO_2$, as well as strong solar radiation. The intensive solar radiation around noon can promote production of hydroxyl (OH) radical, and further oxidizes substantial precursors to form secondary particles. However, the 22:00-00:00 peak is mostly attributed to the other two factors. Firstly, the primary $PM_{2.5}$ obviously is decreased due to the reduction of emission activities at night. Meanwhile, the secondary $PM_{2.5}$ requires some time to generate and accumulate, thus lagging behind changes in emission.

Secondly, nitrate particles can also be produced via $N_2O_5$ heterogeneous reactions in the nighttime.

[Figure]

**Figure R3.** (a) The diurnal distribution of anthropogenic emissions from power plants, industry, residential and transportation (Unit: %). (b) The diurnal variation of the estimated nationwide secondary proportion of $PM_{2.5}$ (Unit: %).

11. Section 4.4: Why did you exclude the wet deposition case here but include in

other sections? I would suggest adding the application condition for your method somewhere.

**Response:** Thank you for your highly careful reminding. Section 4.4 is aimed at discussing the statistical correlation between ozone versus $PM_{2.5}$. We used the daily concentration of these two variables as inputs for further investigation. For $PM_{2.5}$, the 24-h mean concentration can be applied to representing its daily level. The maximum daily 8-h average ozone concentration (MDA8) is usually adopted for describing its concentration level on the daily time-scale. As the reviewer said in the 10th point, ozone is a kind of typical secondary air pollutant with distinctive diurnal pattern (Wang et al., 2017). As shown in Fig. R3, the precipitation process can destroy this diurnal pattern because of the extremely weak radiative condition on rainy days. Meanwhile, ozone concentration level under this condition is mainly affected by background fields. Therefore, MDA8 of rainy days can reveal the background concentration characteristics but not the intensity of secondary formation. To explain the relationship between PM and $O_3$ from the aspect of chemical generation, removing the background dominated cases of $O_3$ concentrations which under precipitation is necessary. We have followed your suggestion to add the explanation for using this preprocessing and rephrase the related texts in Section 4.4.

***Revision in Section 4.4:***

*The $O_3$ diurnal formation regime can be destroyed because of the suppressed radiative condition under precipitation. The local $O_3$ concentration level is mainly dominated by background fields. Here we would like to focus our attention on the secondary formation relationship between daily $PM_{2.5}$ and $O_3$. Therefore the cases when precipitation took place were removed to avoid the cleaning impacts of wet deposition on MDA8 (maximum daily 8-h average) $O_3$ concentrations.*

[Figure]

**Figure R3.** The diurnal variations of $O_3$ concentration in Shanghai on 11 Mar (sunny weather) and 17 Mar (rainy weather), 2022 based on the observations from MEE.

12. The general method to calculate the portion of secondary $PM_{2.5}$ is chemical transport model using bottom-up inventory. It's better to examine the difference in the result between your method and CTM with same inventory.

**Response:** Thanks for your highly conducive comments and rigorous attitude to scientific research. It is really an awesome suggestion. We completely agree that chemical transport model (CTM) is another useful tool to reveal the aerosol compositions. It is interesting to conduct a parallel comparison between two kinds of modeling methods. To examine the difference in result between the MTEA approach and traditional CTM, we adopted the monthly simulated PPM/SPM concentrations from a data fusion system developed by Tsinghua University. This system, which is named Tracking Air Pollution in China (TAP), integrates ground measurements, satellite remote sensing retrievals, emission inventories (MEIC), and CTM simulations (WRF/CMAQ) based on machine learning algorithms. More descriptions of this dataset can be found at http://tapdata.org.cn/ (Geng et al., 2021; Geng et al., 2017). We treated the PPM and SPM concentrations from TAP as the state-of-the-art model representation. Then we showed comparisons between MTEA and TAP in terms of PPM, SPM concentrations and their annual trends in 31 populous cities of China (Fig. R4). In general, comparisons indicate that MTEA estimation has a good agreement with the CTM simulation. To add this part in the manuscript suggested by reviewer, we introduced the TAP dataset in Section 2.3 and showed the related comparisons in Section 3.1.3.

***Revision in Section 2.3:***

***2.3 PPM and SPM estimated by CTM***

[revised manuscript text omitted]

Li, N., Tang, K., Wang, Y., Wang, J., Feng, W., Zhang, H., Liao, H., Hu, J., Long, X.,
Shi, C., and Su, X.: Is the efficacy of satellite-based inversion of $SO_2$ emission
model dependent?, Environmental Research Letters, 16, 10.1088/1748-
9326/abe829, 2021.

Wang, T., Xue, L., Brimblecombe, P., Lam, Y. F., Li, L., and Zhang, L.: Ozone
pollution in China: A review of concentrations, meteorological influences,
chemical precursors, and effects, Sci. Total Environ., 575, 1582-1596,
10.1016/j.scitotenv.2016.10.081, 2017.

Zhang, H., Tang, K., Feng, W., Yan, X., Liao, H., and Li, N.: Impact of Short-Term
Emission Control Measures on Air Quality in Nanjing During the Jiangsu
Development Summit, Frontiers in Environmental Science, 9,
10.3389/fenvs.2021.693513, 2021.

Zheng, B., Zhang, Q., Geng, G., Chen, C., Shi, Q., Cui, M., Lei, Y., and He, K.:
Changes in China's anthropogenic emissions and air quality during the COVID-
19 pandemic in 2020, Earth System Science Data, 13, 2895-2907, 10.5194/essd-
13-2895-2021, 2021.

---

## Author Comment (AC2)

**Response to RC#2:**

Dear Editor and anonymous referee #1:

We greatly appreciate your consideration and the reviewer's constructive comments on the manuscript of "Estimation of Secondary $PM_{2.5}$ in China and the United States using a Multi-Tracer Approach" (acp-2021-683). We have carefully revised the manuscript to address all the comments as described below. Reviewer comments are shown in blue. Our responses are shown in black. The revised texts are shown in italics.

The manuscript demonstrates the multi-tracer estimation algorithm (MTEA), to identify the primary and secondary components from routine observation of $PM_{2.5}$ and validates the method by comparing the long-term and short-term measurements of aerosol chemical composition in China and a network from the United States. This method provides a useful and uncomplicated way to estimate primary and secondary PM, using routine observation species and emission inventories. This manuscript aims to address important questions quantifying primary and secondary aerosols and is within the scope of ACP.

However, regarding the method itself, the method should be carefully introduced with more details. The validation part is a bit weak and should be strengthened in the next version. It is vital because only with good validation can one trust the result from the model. In addition, in the result and discussion part, the discussion is superficial, which needs to be improved in depth, and backed up by more scientific evidence and/or publications.

As a conclusion, the manuscript provides a novel algorithm in primary and secondary particle concentrations, however, the manuscript is not carefully written from the perspective of science and scientific writing, with certain degree of improvement for publication in ACP. Therefore, this manuscript needs a major revision in terms of major context and English language.

**Response:** We thank the reviewer for the comments. According to the reviewer's helpful and insightful comments, we have revised our manuscript and the point-by-point responses to the specific comments were given subsequently. We sincerely hope these revisions are able to address the reviewer's concerns.

1. Introduction: the introduction is poorly written and need to be re-write. If I were you, I would write the introduction based on this outline: 1) introduction of atmospheric aerosols, including sources, type, chemical composition and impacts on air quality, human health and climate, 2) summaries other studies, you must state what has been achieved and what is the current challenging, 3) what is your paper about, how this paper can narrow the gap.

In the current version, the point 1) is addressed, but should be introduced in smoother way. The author is trying to address the point 2), but the studies mentioned in the paragraph 3 in page 3 look not very relevant. For example, the author summarizes the online and offline studies, which is good, and people can see the drawbacks of field and lab measurement to study the PPM and SPM, so the next paragraph should state to overcome these drawbacks, people use model to study the PPM and SPM, and should also state what these model studies have achieved and/or the drawbacks of these method. Finally, this paragraph can lead the final paragraph in the introduction, namely, introduce this study and how this study advances the model studies on PPM and SPM estimation.

**Response:** Thanks for your constructive suggestions and rigorous attitude to scientific research. We do think it is necessary and important to rephrase the structure of this part. Following the suggestion, we have rewritten the introduction section. The detailed description of this part has been corrected in the revised manuscript as follows.

***Revision in Section 1:***

*Fine particulate matter ($PM_{2.5}$, aerodynamic diameter less than 2.5 μm) can be categorized into primary and secondary $PM_{2.5}$ according to its formation processes.*

*Primary PM$_{2.5}$ (PPM), including primary organic aerosol (POA), elemental carbon (EC), sea salt and mineral dust, is the product of direct emission from combustion of fossil/biomass fuel, dust blowing and sea spray. Secondary PM$_{2.5}$ (SPM) mainly generates from the further oxidation of gaseous precursors emitted by anthropogenic and biogenic activities (Zhu et al., 2018; Wang et al., 2019). SPM consists of secondary organic aerosol (SOA) and secondary inorganic aerosol (SIA, including sulfate, nitrate and ammonium). The primary and secondary components of PM$_{2.5}$ have different environmental impacts on air quality, human health and climate change. For example, as a typical PPM, EC can severely reduce atmospheric visibility and greatly influence weather and climate due to its strong absorption of solar radiation (Bond et al., 2013; IPCC, 2013; Mao et al., 2017). Sulfate, a critical hygroscopic component of secondary PM$_{2.5}$ (SPM), can be fast formed under high relative humidity conditions and further leads to grievous air pollution (Cheng et al., 2016; Guo et al., 2014; Quan et al., 2015). Furthermore, the sulfate and other hygroscopic PM$_{2.5}$ have considerable influences on climate change mostly by changing cloud properties (Leng et al., 2013; von Schneidemesser et al., 2015). In addition, different PM$_{2.5}$ components also have various deleterious impacts on human health for their toxicities (Hu et al., 2017; Khan et al., 2016; Maji et al., 2018).*

*To understand the severe PM$_{2.5}$ pollution characteristics in China over the past several years (An et al., 2019; Song et al., 2017; Yang et al., 2016), many observational studies have been conducted on PM$_{2.5}$ components. The basic methods of these studies are offline laboratory analysis and online instrument measurement such as aerosol mass spectrometer (AMS). The observational studies are crucial to exactly identify the aerosol chemical compositions. For offline approach, it is the most widely used method (Ming et al., 2017; Tang et al., 2017; Tao et al., 2017; Dai et al., 2018; Gao et al., 2018; Liu et al., 2018a; Wang et al., 2018; Zhang et al., 2018; Xu et al., 2019; Yu et al., 2019) and is successfully applied to investigate the inter-annual variations of different aerosol chemical species (Ding et al., 2019; Liu et al., 2018b). In terms of online approach, AMS is the state-of-the-art method for analyzing different chemical species with high time resolution, which has great application value in diagnosing the causes of haze events in China over the past decade (Huang et al., 2014; Quan et al., 2015; Guo et al., 2014; Yang et al., 2021; Gao et al., 2021; Hu et al., 2021; Zhang et al., 2022).*

*Nevertheless, both the online and offline measurements require a high level of*

*manpower and economic cost, and for this reason, these methods are expensive and rarely applied in large-scale regions or long-term periods.*

*Chemical transport model (CTM) is another useful tool to identify the composition characteristics of $PM_{2.5}$. The simulation predicted by CTM is featured as high spatio-temporal resolution (Geng et al., 2021). Meanwhile, it also provides vertical profiles of diverse chemical species (Ding et al., 2016). However, the CTM results are largely dependent on external inputs such as emission inventories, boundary conditions, initial conditions, etc. The internal parameterizations of itself significantly influence the final model results as well (Huang et al., 2021), which leads to uncertainty in the simulated $PM_{2.5}$ and its composition. In addition, the burden of high requirement in computational cost and storage also makes CTM hard to universally use.*

*In this study, we develop a novel method, Multi-Tracer Estimation Algorithm (MTEA), with the aim of distinguishing the primary and secondary compositions of $PM_{2.5}$ from routine observation of $PM_{2.5}$ concentration. Different from traditional CTMs, MTEA proposed by this study is based on statistical assumption and works in a more convenient way. This algorithm and its application are tested in China and the United States. In Section 2, we introduce the structure and principle of MTEA. In Section 3, we evaluate the MTEA results comparing with three $PM_{2.5}$ composition data sets, (1) short-term measurements in 16 cities in China from 2012 to 2016 reported by previous studies, (2) continuous long-term measurements in Beijing and Shanghai from 2014 to 2018, and (3) IMPROVE network in the United States during 2014 and 2018. Additionally, we also compare MTEA model with one of the most advanced datasets from CTM in China. Subsequently, in Section 4 we investigate the spatio-temporal characteristics of PPM and SPM concentrations in China, explain the unexpected haze event in several cities of China during the COVID-19 lockdown and discuss the complicated correlation between PM and $O_3$. This study is different from previous works as follows: (1) we develop an efficient approach to explore PPM and SPM with low economy-/technique-cost and computation burden, (2) we apply this approach to observation data from the MEE network, offering an unprecedented opportunity to quantify the $PM_{2.5}$ components on a large space and time scale.*

2. Methodology: the methodology part is written in a reasonable logic, but the author needs to pay more attention to specify the technical details, e.g., the definition of some terms.

**Response:** Thanks for your kind reminding and rigorous attitude to scientific research. We have carefully checked all technical details and revised them for a more proper expression in Section 2.

***Revision in Section 2:***

*The multi-tracer (marked as X) is defined to represent multiple primary contributions to $PM_{2.5}$, mainly resulting from incomplete combustion of carbonaceous material and flying dust.*

*We select the typical combustion product CO as one tracer to represent the combustion process, and the particles in coarse mode ($PM_{coarse}$, marked as PMC, PMC = $PM_{10}$ – $PM_{2.5}$) as the other tracer to track flying dust.*

*However, this investigative coefficient for quantifying primary sulfate and nitrate emissions might be relatively higher compared to empirical coefficients (0.01-0.05) used in previous simulation studies.*

*They estimated primary and secondary organic carbon (marked as POC and SOC) concentrations by adopting a proper POC/EC ratio when SOC correlated with EC worst.*

3. Model validation: this part straightforwardly delivers the good validation result between model and observation. Good correlation is shown in this part, suggesting good model performance. However, this part also requires more interpretation on the model's over/underestimation behavior compared to observation, which is now absent. Ideally, the author should focus most on this part, because only when the model is reasonable validated can we trust the result and make the further interpretation on the result.

Therefore, from my own perspective, the author should strengthen this part.

**Response:** Thanks for your conducive comment. We have enhanced the discussion in the model evaluation part as you suggested.

***Revision in Section 3.1.2:***

*However, we find that there are still a few discrepancies between the estimated and*

*observation-based results. For example, we overestimated the secondary proportions*

*of $PM_{2.5}$ in cities such as Haikou, Lanzhou and Lhasa. Though all of them show a*

*considerable overestimation of over 20%, the causes lead to this kind of bias may be*

*quite different. In coastal city Haikou, we may attribute this discrepancy between MTEA*

*and observation to the neglect of the contribution of sea salt aerosols. The $PM_{2.5}$ offline*

*measurements in 2015 exhibited that the contribution of sea salt aerosols to ambient*

*$PM_{2.5}$ mass concentration in Haikou is 3.6-8.3% (Liu et al., 2017). Secondly, the*

*overestimation phenomenon in Lanzhou, which is a typical inland city located in*

*northwestern China, can be explained by overlooking the contribution of natural dust*

*to $PM_{2.5}$ speciation. Generally, both sea salt and natural dust are categorized into non-*

*anthropogenic processes, and are not accounted for by anthropogenic emission*

*inventory, resulting in the underestimation of representing primary process intensity.*

*Finally, for Lhasa, the observation-based results which are derived from too few*

*samplers also pose controversial comparison against MTEA model.*

4. Result and discussion: this part also very straightforwardly and logically reports the results. However, the interpretation of results should be more comprehensive and backed up by previous studies and/or solid evidence, which is absent now and needs to be added. In addition, the discussion of the result is very superficial, lacking depths, which should also be improved.

**Response:** Thanks for your conducive comments and rigorous attitude to scientific research. To enrich our discussion as the reviewer mentioned, we have carefully revised the related texts in the result part.

***Revision in Section 4:***

*We used the MTEA approach and the MEE observation data to estimate PPM and*

*SPM concentrations in China for the period of 2014-2018. The observations during*

*severe haze events (top 10% CO and PMC polluted days) were excluded to avoid the*

*influence of unfavorable meteorological conditions and extreme high primary emission*

*cases. Unfavorable meteorological conditions are major causes for haze events. PPM*

*under these unfavored meteorological conditions may have considerable high co-linear*

*relationship with total $PM_{2.5}$. The concentration of SPM from complicated formation*

*pathways is then underestimated. Therefore, we excluded these polluted days to focus*

*more attention on general characteristics of PPM and SPM concentration.*

***Revision in Section 4.3:***

*To explore this unexpected air pollution, we find that the enhanced secondary*

*pollution could be the major factor, which even offset the reduction of primary*

*emissions in the BTH region during the lockdown. With the help of MTEA, we tracked*

*variations of the secondary proportions of $PM_{2.5}$ in East China before and during the*

*COVID-19 lockdown (Fig. 9 d-f). The specific emission reductions owing to the*

*national lockdown were derived from Huang et al. (2020). Based on the bottom-up*

*dynamic estimation, provincial emissions of CO, $NO_x$, $SO_2$, VOC, $PM_{2.5}$, BC and OC*

*decreased by 13-41%, 29-57%, 15-42%, 28-46%, 9-34%, 13-54%, and 3-42%,*

*respectively during the lockdown period. The secondary proportions in the BTH region*

*show an evident increase, at the level of 7%-34%, which highlights the importance of*

*the secondary formation during the lockdown. Our result is consistent with recent*

*observation and simulation studies (Chang et al., 2020; Huang et al., 2020; Le et al.,*

*2020), which suggested that the reduced $NO_2$ resulted in $O_3$ enhancement, further*

*increasing the AOC and facilitating the formation of secondary inorganic aerosols such*

*as ammonium sulfate, ammonium nitrate. In addition, another cause of the air pollution*

*is the unfavorable atmospheric diffusion conditions. CO, a nonreactive pollutant, was*

*increased by 22% in Beijing during the lockdown even under considerable reduction*

*on its emission.*

*Revision in Section 4.4:*

*A series of recent studies have focused on the correlation between $PM_{2.5}$ and $O_3$, and many of them agreed that the correlation varies greatly in different regions of China. Specifically, the statistical correlation is stronger positive in southern cities compared to that in northern cities (Chu et al., 2020). Because of this significant difference, a question raises: is the difference mostly caused by PPM, or SPM, or both of them? To address this question, we compare the correlations between daily PPM, SPM and total $PM_{2.5}$ versus $O_3$ in Beijing-Tianjin-Hebei (BTH) and Yangtze River Delta (YRD) region during the study period, with the help of META approach. The $O_3$ diurnal formation regime can be destroyed because of the suppressed radiative condition under precipitation. The local $O_3$ concentration level is mainly dominated by background fields. Here we would like to focus our attention on the secondary formation relationship between daily $PM_{2.5}$ and $O_3$. Therefore the cases when precipitation took place were removed to avoid the cleaning impacts of wet deposition on MDA8 (maximum daily 8-h average) $O_3$ concentrations. Precipitation data is based on the ERA5 reanalysis database from the European Centre for Medium-Range Weather Forecasts (ECMWF, https://www.ecmwf.int/, last access, 1 August 2021).*

*Revision in Section 4.5:*

*Thirdly, current bottom-up emission inventories are generally outdated with a time lag of at least 1-2 years, mainly due to the lack of timely and accurate statistics. Consequently, the adjoint uncertainty in MTEA estimation is inevitable.*

*To evaluate the uncertainty, a comparison test was conducted by adjusting the apportioning coefficient (the a and b in Eq. 1) with a disturbance of ±0.1. Firstly, we decreased the value of a in each populous city by 0.1. Meanwhile, the coefficient b increased by 0.1. This scenario indicates an overestimation in contribution of combustion-related process to primary $PM_{2.5}$ or underestimation in contribution of dust-related process. Secondly, we increased the value of a in each populous city by 0.1 (decreased b by 0.1) for checking the opposite case. The results are presented in Table*

*S5 and point out that the estimated secondary proportions of PM$_{2.5}$ varied less than ±3% in most populous cities caused by the changes of the apportioning coefficient. This sensitivity experiment highlights that the apportioning coefficients depending on emissions has limited impacts on the final estimation results. Generally, the uncertainty of apportioning coefficient is one of two factors that directly affect the tracer X. The other one is the concentration of CO and PMC itself. Hence, we also conducted a similar test to check the impacts of tracer X on the model estimation by changing the tracer concentrations mentioned in Eq.1. Specifically, we (1) increased CO concentration by 10% as well as decreased PMC concentration by 10% and (2) decreased CO concentration by 10% as well as increased PMC concentration by 10%. Both sets of adjustment show changes within ±2% in the estimated secondary proportions of PM$_{2.5}$ in all cities except for Urumqi (Table S6). This phenomenon from the perspective of tracer concentration also supports that the impacts of the tracer X on the final model results are limited. In summary, we believe that the most determinative stuff for the final results of our model is the principle of the minimum correlation between PPM and SPM but not the tracer X which relies on emissions or concentrations.*

5. Conclusion: it summarizes the significance of the study, but one or two paragraph need to be re-written, based on the revised context in Section 4.

**Response:** Thank you for your comments and we have added the related texts to the manuscript.

***Revision in Section 5:***

[revised manuscript text omitted]

---

## Author Comment (AC3)

**Response to RC#3:**

Dear Editor and anonymous referee #4:

We greatly appreciate your consideration and the reviewer's constructive
comments on the manuscript of "Estimation of Secondary PM$_{2.5}$ in China and the
United States using a Multi-Tracer Approach" (acp-2021-683). We have carefully
revised the manuscript to address all the comments as described below. Reviewer
comments are shown in blue. Our responses are shown in black. The revised texts are
shown in italics.

1. The manuscript presents a method for estimating the relative contributions of
primary and secondary PM by proxy. The observed input parameters are CO, PM$_{10}$ and
PM$_{2.5}$, however, the method also relies on estimated emissions of OA, EC, OC, fine
dust, PM$_{2.5}$, sulfate and nitrate, from emission inventories. The authors develop a proxy
for secondary particulate matter on the basis of the observed parameters and estimated
emissions. The motivation is presented as the need for a low cost, operational method
for monitoring the contributions of secondary aerosols to the total PM$_{2.5}$ levels.

The method appears to have some use for informing operational air quality
management or for informing policy, but the scientific value of the method is not
convincingly presented. It relies on assumptions and inventories that are not universal,
and the manuscript does not present a convincing argument for its use, other than that
it is cheaper than source apportionment methods based on chemical speciation. But it
does not present comparative estimates of primary-secondary contributions with those
methods.

It is questionable if this method has any value. It requires a big body of inputs, as
other chemical transport models, but also relies heavily on assumptions and coefficients
that are externally adjusted, even tuned to fit the model.

**Response:** Thank you for the comments. The traditional methods to identify

PM$_{2.5}$ compositions include observational and simulation methods. The observational method is currently the most common and useful way for quantitatively investigating the PM$_{2.5}$ chemical compositions. Moreover, chemical transport model (CTM) is another useful tool to identify the composition characteristics of PM$_{2.5}$. However, the

CTM results are largely dependent on external inputs as the reviewer mentioned such as emission inventories, boundary conditions, initial conditions, etc. The internal parameterizations of itself significantly influence the final model results as well (Huang et al., 2021).

Different from CTM, the MTEA model developed in this study is a statistical model, which does not suffer from the burden of high requirement in computational cost and storage. MTEA is positioned as a low economy-/technique-cost tool to conveniently estimate the primary and secondary PM$_{2.5}$ in both scientific and practical areas, although concomitantly it is slightly inferior to the two traditional methods in terms of identifying detailed PM$_{2.5}$ compositions and capturing high temporal variation.

The aim of this study, by using MTEA, is to reveal the general characteristics of primary and secondary PM$_{2.5}$ pollution over wide spatio-temporal coverages. The evaluation between MTEA estimation versus various measurements in terms of monthly mean value shows a satisfying performance (Section 3). At the same time, the reasonable spatio-temporal patterns of PPM and SPM concentrations disclosed by our model also inform that MTEA is a promising tool for illustrating general pollution patterns. Thus, for studies which would like to distinguish primary and secondary PM$_{2.5}$,

MTEA model can serve as a potential option. In the future, we also hope to cooperate with the team which focuses on observational studies to broaden the application of

MTEA and reduce the uncertainty. Thanks again and we have rephrased our texts in the manuscript for a clearer description in terms of the scientific value of our method in

Section 1.

***Revision in Section 1:***

*Nevertheless, both the online and offline measurements require a high level of*

*manpower and economic cost, and for this reason, these methods are expensive and rarely applied in large-scale regions or long-term periods.*

*Chemical transport model (CTM) is another useful tool to identify the composition characteristics of $PM_{2.5}$. The simulation predicted by CTM is featured as high spatio-temporal resolution (Geng et al., 2021). Meanwhile, it also provides vertical profiles of diverse chemical species (Ding et al., 2016). However, the CTM results are largely dependent on external inputs such as emission inventories, boundary conditions, initial conditions, etc. The internal parameterizations of itself significantly influence the final model results as well (Huang et al., 2021), which leads to uncertainty in the simulated $PM_{2.5}$ and its composition. In addition, the burden of high requirement in computational cost and storage also makes CTM hard to universally use.*

*In this study, we develop a novel method, Multi-Tracer Estimation Algorithm (MTEA), with the aim of distinguishing the primary and secondary compositions of $PM_{2.5}$ from routine observation of $PM_{2.5}$ concentration. Different from traditional CTMs, MTEA proposed by this study is based on statistical assumption and works in a more convenient way. This algorithm and its application are tested in China and the United States. In Section 2, we introduce the structure and principle of MTEA. In Section 3, we evaluate the MTEA results comparing with three $PM_{2.5}$ composition data sets, (1) short-term measurements in 16 cities in China from 2012 to 2016 reported by previous studies, (2) continuous long-term measurements in Beijing and Shanghai from 2014 to 2018, and (3) IMPROVE network in the United States during 2014 and 2018. Additionally, we also compare MTEA model with one of the most advanced datasets from CTM in China. Subsequently, in Section 4 we investigate the spatio-temporal characteristics of PPM and SPM concentrations in China, explain the unexpected haze event in several cities of China during the COVID-19 lockdown and discuss the complicated correlation between PM and $O_3$. This study is different from previous works as follows: (1) we develop an efficient approach to explore PPM and SPM with low economy-/technique-cost and computation burden, (2) we apply this approach to observation data from the MEE network, offering an unprecedented opportunity to quantify the $PM_{2.5}$ components on a large space and time scale.*

2. The manuscript describes comparisons between estimated and observed primary particulate matter. Categorization of measured historical data into secondary and primary aerosols for comparison with the MTEA seems to be based on chemical compositions, but this process is not clearly described and the criteria are vague. There has been no attempt to verify the MTEA estimates for ppm by comparing with published estimates based on receptor modelling, CTMs or AMS studies. There are many studies in the literature that have produced estimates that can be easily compared with the outcomes of the MTEA approach, but that has not been done.

**Response:** Thank you for pointing this out. We have added the description about categorizing the concentrations of measured historical aerosol chemical species into PPM and SPM concentrations in Section 2.2.2 and 2.2.3.

The estimation from MTEA model is based on the routine $PM_{2.5}$ observation. However, the measurements from literature we summarized in Section 3.1.2 rely on sampling at different locations. The measurements may be quite different though the observational campaigns were conducted in the same city. Thus it is difficult to directly compare PPM concentrations predicted by MTEA with that in various literature. Therefore, we mainly focus on the comparison in terms of secondary proportions of $PM_{2.5}$ between the MTEA method versus various previous studies. Please refer to Table S4 in the supplementary material for the specific comparisons. Moreover, we also revised Table S4 to clearly show the method applied by these previous studies (offline sampling or AMS instrument).

To examine the difference in result between the MTEA approach and traditional CTM, we adopted the monthly simulated PPM/SPM concentrations from a data fusion system developed by Tsinghua University. This system, which is named Tracking Air Pollution in China (TAP), integrates ground measurements, satellite remote sensing retrievals, emission inventories (MEIC), and CTM simulations (WRF/CMAQ) based on machine learning algorithms. More descriptions of this dataset can be found at http://tapdata.org.cn/ (Geng et al., 2021; Geng et al., 2017). We treated the PPM and SPM concentrations from TAP as a typical model representation. To add this part in the manuscript suggested by reviewer, we introduced TAP dataset in Section 2.3 and showed comparisons between MTEA and TAP in terms of PPM, SPM concentrations as well as their annual trends in 31 populous cities of China in Section 3.1.3.

***Revision in Section 2.2.2:***

*After accessing the chemical compositions, we categorized them into PPM and SPM for further evaluation. Specifically, SOA was roughly identified from OM by EC-tracer model (Ge et al., 2017). SPM concentrations were calculated via summing $SO_4^{2-}$, $NO_3^-$, $NH_4^+$ and SOA concentrations. Then PPM could be calculated though deducting SPM from $PM_{2.5}$.*

*In addition, we investigated observation-based $PM_{2.5}$ component analyses in 16 cities of China during 2012-2016 from 32 published studies. This survey offered an opportunity to compare the estimation by MTEA with the past measurements in the terms of the secondary fraction of $PM_{2.5}$. SPM concentrations in literature were roughly estimated by multiplying OM from 0.5 because of the limit of data source. Meanwhile, it is noted that the factor which converts OC to OM is dependent on the definition of each observation study itself.*

***Revision in Section 2.2.3:***

*The specific aerosol chemical compositions include ammonium sulfate, ammonium nitrate, organic/elemental carbon and soil/mineral dust. The categorization for PPM and SPM in IMPROVE dataset is similar to the process in Section 2.2.2. The only difference is that SPM concentration is the sum of ammonium sulfate, ammonium nitrate and SOA.*

***Revision in Section Table S4:***

***#Please see below#***

**Table S4.** List of $PM_{2.5}$ component measurements ($\mu g\ m^{-3}$) of China in previous studies.

[revised manuscript text omitted]

3. It is true as the authors state that those other methods are labor-intensive and expensive, but they are also scientifically tried and tested and therefore more convincing, so it would make sense to develop the performance of the MTEA against such methods more than has been done in this manuscript.

**Response:** Thank the reviewer for pointing this out. There is no doubt that the measurements via offline or online methods are absolutely crucial to scientifically understanding the compositions of $PM_{2.5}$. To some extent, the identification of $PM_{2.5}$ based on these methods offers a conclusive insight for model developers, and the MTEA model we developed should be in line with the observational results. We heartfeltly acknowledged the efforts that the highly scientific observations made. We compared the MTEA results with a series of observational studies as shown in Table S4, and revised the related text in Section 2.2.

In addition, this study mainly devotes to illustrating the general pattern of primary and secondary $PM_{2.5}$ pollution over a wide spatio-temporal coverage with the aid of a convenient proxy tool, and has no intention to replace the crucial observational methods with MTEA. Thank you for the review's comment again and we have revised the related texts in Section 1 to clarify the roles and relationships between observational method and MTEA.

***Revision in Section 1:***

*To understand the severe $PM_{2.5}$ pollution characteristics in China over the past several years (An et al., 2019; Song et al., 2017; Yang et al., 2016), many observational*

*studies have been conducted on PM$_{2.5}$ components. The basic methods of these studies are offline laboratory analysis and online instrument measurement such as aerosol mass spectrometer (AMS). The observational studies are crucial to exactly identify the aerosol chemical compositions. For offline approach, it is the most widely used method (Ming et al., 2017; Tang et al., 2017; Tao et al., 2017; Dai et al., 2018; Gao et al., 2018; Liu et al., 2018a; Wang et al., 2018; Zhang et al., 2018; Xu et al., 2019; Yu et al., 2019) and is successfully applied to investigate the inter-annual variations of different aerosol chemical species (Ding et al., 2019; Liu et al., 2018b). In terms of online approach, AMS is the state-of-the-art method for analyzing different chemical species with high time resolution, which has great application value in diagnosing the causes of haze events in China over the past decade (Huang et al., 2014; Quan et al., 2015; Guo et al., 2014; Yang et al., 2021; Gao et al., 2021; Hu et al., 2021; Zhang et al., 2022).*

4. Also, the manuscript states that the numerical calculations were done on a supercomputing system. It can be argued that if the approach requires a supercomputing facility, then it is no less costly or inaccessible than the existing source apportionment methods, but the cost has been shifted from scientific equipment to IT services.

**Response:** Thank you for the comment and the careful reminding. We indeed agree that traditional numerical models such as WRF-Chem, CMAQ and CAMx does cost considerable computational sources. However, our model is based on the statistical principle. Actually, it is capable of running on the personal computer (PC) platform with basic equipment requirement. In the future, we look forward to simplifying the model for a more lightweight version so that it can be easily utilized for application anywhere.

5. The manuscript does touch on a discussion that has scientific interest, and that is contained in the sections 4.1 and 4.2 on spatial and temporal variation. The discussion on spatial variation has some merit. There is potentially a better motivation for developing the MTEA approach in order to inform a discussion on the spatial and temporal variation where only proxy parameters are available, by leveraging national monitoring networks to learn more about geographical distribution of secondary aerosols and feed into a discussion on variations in atmospheric processes.

**Response:** Thank you for your comments and rigorous attitude to scientific research. We have rephrased our statement of the motivation for this study in Section 5.

***Revision in Section 5:***

[revised manuscript text omitted]